# Plug-and-Play Benchmarking of Reinforcement Learning Algorithms for Large-Scale Flow Control

**Jannis Becktepe** [1 2]  **Aleksandra Franz** [3 4]  **Nils Thuerey** [3 4]  **Sebastian Peitz** [1 2]

## Abstract

Reinforcement learning (RL) has shown promising results in active flow control (AFC), yet progress in the field remains difficult to assess as existing studies rely on heterogeneous observation and actuation schemes, numerical setups, and evaluation protocols. Current AFC benchmarks attempt to address these issues but heavily rely on external computational fluid dynamics (CFD) solvers, are not fully differentiable, and provide limited 3D and multi-agent support. To overcome these limitations, we introduce FluidGym, the first standalone, fully differentiable benchmark suite for RL in AFC. Built entirely in PyTorch on top of the GPU-accelerated PICT solver, FluidGym runs in a single Python stack, requires no external CFD software, and provides standardized evaluation protocols. We present baseline results with PPO, SAC, DPC, and TD-MPC, and release all environments, datasets, and trained models as public resources. FluidGym enables systematic comparison of control methods, establishes a scalable foundation for future research in learning-based flow control, and is available at github.com/safe-autonomous-systems/fluidgym.

## 1. Introduction

Active flow control (AFC) plays a central role in a wide range of real-world systems, such as aerodynamics (Batikh et al., 2017), energy harvesting (Barthelmie et al., 2009), nuclear fusion (Pironti & Walker, 2005), and reduction of turbulence (Jiménez, 2013). Europe, for instance, could save

[1]TU Dortmund University, Dortmund, Germany [2]Lamarr Institute for Machine Learning and Artificial Intelligence, Dortmund, Germany [3]Technical University Munich, Munich, Germany [4]Munich Center for Machine Learning, Munich, Germany. Correspondence to: Jannis Becktepe <jannis.becktepe@tu-dortmund.de>.

*Proceedings of the 43rd International Conference on Machine Learning*, Seoul, South Korea. PMLR 306, 2026. Copyright 2026 by the author(s).

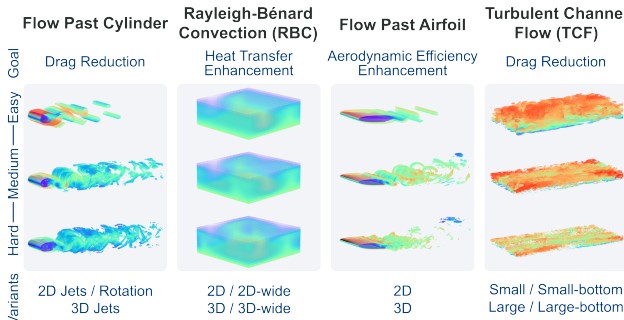

*Figure 1.* The four uncontrolled environment classes in FluidGym.

more than $20 \times 10^6$ tonnes of $CO_2$ per year by reducing drag on cars using AFC (Brunton & Noack, 2015).

However, manually designing control strategies is challenging due to the high dimensionality and inherent nonlinearities of such systems (Duriez et al., 2017). Recently, reinforcement learning (RL) has demonstrated strong potential for advancing AFC in complex systems, e.g., stabilizing the plasma in a Tokamak reactor (Degrave et al., 2022).

Despite its success, research in RL for flow control remains fragmented, and establishing a clear state of the art is difficult for several reasons. Experimental setups vary widely across studies in terms of actuators, sensor placements, and physical parameter settings as well as RL algorithms and hyperparameters (Viquerat et al., 2022; Moslem et al., 2025). This results in inconsistent problem formulations that hinder direct comparisons. Moreover, insufficiently rigorous evaluation and the use of few random seeds increase statistical variance (Henderson et al., 2018; Agarwal et al., 2021).

Existing benchmarks (see Table 1) have seen limited adoption for two main reasons. First, most rely on external computational fluid dynamics (CFD) solvers that must be installed, configured, and coupled to Python RL code through additional interfaces, which demands CFD expertise and creates brittle software stacks. Second, differentiability is either absent or limited to a small subset of scenarios, which prevents end-to-end use of Differentiable Predictive Control (DPC; Drgoňa et al. (2022)) and recent differentiable RL methods (Xing et al., 2025; Lagemann et al., 2025b).

To address these limitations, we introduce FluidGym, the first standalone, fully differentiable RL benchmark for AFC in incompressible flows. Building entirely on PyTorch (Ansel et al., 2024), FluidGym requires no external solver dependencies and seamlessly integrates with common RL interfaces such as Gymnasium (Towers et al., 2024) or PettingZoo (Terry et al., 2021) and algorithm frameworks like Stable-Baselines3 (Raffin et al., 2021) or TorchRL (Bou et al., 2023). As all simulations and control interfaces live in one Python package, users can install FluidGym via `pip` and immediately run experiments with standard RL libraries, without compiling or coupling external CFD codes. Being inherently end-to-end differentiable, FluidGym enables researchers to use gradient-based control methods alongside classical RL without any further modifications. Our benchmark provides diverse environments with consistent task definitions, supports single-agent RL (SARL) and multi-agent RL (MARL) settings, spans three difficulty levels in 2D and 3D, and enables transfer-learning studies.

In summary, our main contributions are (1) the first standalone, fully differentiable, plug-and-play benchmark for RL in AFC, implemented in a single PyTorch codebase without external solver dependencies; (2) a collection of standardized environment configurations spanning diverse 3D and MARL control tasks (see Figure 1); (3) an extensive experimental study covering all core FluidGym environment classes and difficulty levels, including transfer-learning evaluations, amounting to over 25 k GPU hours, all publicly available.

## 2. Background and Related Work

**RL for AFC**   Fluid flows are governed by the Navier-Stokes equations, a set of nonlinear partial differential equations (PDEs) exhibiting highly complex behavior over a wide range of scales both in space and time. Due to their inherent complexity, analytical solutions are infeasible without substantial simplifications. Computational fluid dynamics (CFD) has become a standard approach to approximate solutions using spatial and temporal discretization (Ferziger et al., 2020). Such simulations, however, are computationally expensive and typically require specialized solvers such as OpenFOAM (Weller et al., 1998), FEniCS (Alnæs et al., 2015), or FLEXI (Krais et al., 2021).

In many applications, the goal is not only to simulate the flow but to *manipulate* it. Active flow control (AFC) uses actuation to influence fluid motion, e.g., to reduce aerodynamic drag (Nair et al., 2019). Classical AFC approaches have demonstrated notable successes, ranging from the re-laminarization of turbulent channel flows using adjoint-based model predictive control (MPC) (Bewley et al., 2001) to control of the separation bubble behind a bluff body using evolutionary optimization strategies (Gautier et al., 2015).

However, these methods often rely on simplified models or require full-state information and expensive online optimization, which limits their scalability to complex, nonlinear, or high-dimensional flow configurations. Reinforcement learning (see Appendix A for an introduction to the basics and notation) has therefore emerged as a compelling alternative and has been explored across a variety of AFC problems, including drag reduction in bluff-body wakes (Rabault et al., 2019; Tokarev et al., 2020), turbulent channel-flow control (Guastoni et al., 2023), and heat-transfer enhancement (Beintema et al., 2020; Vignon et al., 2023). Notably, Beintema et al. (2020) were the first to apply RL to control Rayleigh-Bénard convection and established the foundation for subsequent studies on heat-transfer (Vignon et al., 2023).

Real-world AFC often involves high-dimensional, spatially distributed actuation (e.g., arrays of jets or heaters). Several works have also studied multi-agent reinforcement learning (MARL, Albrecht et al. (2024)) for wall turbulence modeling (Bae & Koumoutsakos, 2022), heat transfer enhancement (Beintema et al., 2020; Vasanth et al., 2024; Vignon et al., 2023; Markmann et al., 2025), and proposed convolutional RL for distributed control (Peitz et al., 2024). To avoid the high computational cost of CFD simulations, RL has also been used together with surrogate models (Werner & Peitz, 2024; Zolman et al., 2025).

However, the research area faces challenges similar to those observed more broadly in machine learning for PDEs (Mc-Greivy & Hakim, 2024). Evaluation practices vary widely. Many works compare learned policies only to uncontrolled baselines (Tokarev et al., 2020; Ren et al., 2021; Wang et al., 2022b; Vignon et al., 2023; Vasanth et al., 2024; Ren et al., 2024; Zhao et al., 2024; Suárez et al., 2025; Montalà et al., 2025). RL episodes often start from the same initial state (Rabault et al., 2019; Vignon et al., 2023; Ren et al., 2024; Sonoda et al., 2023; Garcia et al., 2025), even though this choice can substantially affect the performance of policies (Guastoni et al., 2023). In several cases, test episodes reuse the same initial conditions used for training (Vignon et al., 2023; Vasanth et al., 2024), making generalization hard to assess. Reproducibility and statistical robustness are also limited: some works report a single run without seeds (Guastoni et al., 2023; Sonoda et al., 2023), despite the fact that this is a known source of variance in RL (Henderson et al., 2018; Agarwal et al., 2021). Finally, although the soft actor-critic (SAC; Haarnoja et al. (2018)) algorithm often outperforms proximal policy optimization (PPO; Schulman et al. (2017)) on nonlinear continuous-control tasks (Abuduweili & Liu, 2023), more than 75% of AFC studies rely on PPO as surveyed by Moslem et al. (2025). More advanced continuous-control methods (Tavakoli et al., 2021; Seyde et al., 2023) remain largely unexplored in AFC despite their potential to advance the field. This motivates the need for systematic benchmarking of control methods.

*Table 1.* Overview of existing RL for AFC benchmarks in terms of external solver dependence, differentiability of all environments, multi-agent RL support, and 3D capabilities.

| BENCHMARK | NO EXTERNAL SOLVER | FULLY DIFFERENTIABLE | MARL | 3D |
|---|:---:|:---:|:---:|:---:|
| DRLinFluids (Wang et al., 2022a) | ✗ | ✗ | ✗ | ✗ |
| drlfoam (Weiner & Geise, 2022) | ✗ | ✗ | ✗ | ✗ |
| DRLFluent (Mao et al., 2023) | ✗ | ✗ | ✗ | ✗ |
| Gym-preCICE (Shams & Elsheikh, 2023) | ✗ | ✗ | ✗ | ✗ |
| Beacon (Viquerat et al., 2024) | ✓ | ✗ | ✗ | ✗ |
| HydroGym (Lagemann et al., 2025b) | ✗ | ✗ | ✓ | ✓ |
| **FluidGym (Ours)** | ✓ | ✓ | ✓ | ✓ |

**RL for AFC Benchmarks** Benchmark design is key to addressing the evaluation and reproducibility challenges outlined above. Notably, unlike conventional RL benchmarks like MuJoCo (Todorov et al., 2012), which involve low-dimensional state spaces and relatively smooth dynamics, AFC tasks are characterized by extremely high dimensions and chaotic dynamics (Brunton & Noack, 2015). Several RL benchmarks for AFC exist, and their characteristics are stated in Table 1. However, existing efforts cover only parts of the AFC landscape and leave important gaps in accessibility, differentiability, RL methodologies, and dimensionality.

General PDE control benchmarks, such as those proposed by Bhan et al. (2024); Zhang et al. (2024); Mouchamps et al. (2026), focus on low-dimensional or non-fluid systems and do not address the complexities of high-dimensional fluid flows. Several frameworks have attempted to bridge the gap between CFD solvers and RL algorithms (Pawar & Maulik, 2021; Kurz et al., 2022; Xiao et al., 2025). However, they introduce additional software layers for the coupling rather than standardized benchmark environments.

DRLinFluids (Wang et al., 2022a) and drlFoam (Weiner & Geise, 2022) interface with OpenFOAM but are limited to 2D cases (e.g., flow past a cylinder or fluidic pinball), while DRLFluent (Mao et al., 2023) couples RL with the commercial solver Fluent (ANSYS Inc., 2026), again focusing on 2D cylinder flows. Gym-preCICE (Shams & Elsheikh, 2023) uses the preCICE coupling library (Chourdakis et al., 2022) and includes a 2D flow past a cylinder. Beacon (Viquerat et al., 2024) offers a purely Python-based environment without external solver dependencies, but remains limited to 2D single-agent scenarios and is not differentiable. HydroGym (Lagemann et al., 2025b;a) provides a collection of 2D and 3D flow scenarios, with individual environments depending on different solver backends: FEniCS for 2D simulations, and m-AIA (Institute of Aerodynamics, 2024) for 3D simulations. Only the two environments based on JAX (Bradbury et al., 2018) are differentiable. While HydroGym provides a valuable platform that couples different CFD backends with RL interfaces, the setup and interaction with complex CFD codes reduce accessibility for non-experts in fluid dynamics.

**Limitations of Existing Benchmarks** Existing AFC benchmarks share several limitations (see Table 1): (i) they typically depend on external CFD solvers (e.g., OpenFOAM, Fluent, FEniCS, m-AIA), which require complex and often brittle software pipelines and indirect coupling layers that hinder integration with Python RL libraries and complicate long-term maintenance; (ii) lack of differentiability, despite its potential for accelerating RL training (Xu et al., 2022; Xing et al., 2025; Lagemann et al., 2025b) and in DPC (Drgoňa et al., 2022), (iii) limited support for multi-agent RL, despite its natural alignment with spatially distributed actuation; and (iv) predominantly 2D environments, which fail to capture essential 3D flow physics. To our knowledge, no existing benchmark simultaneously provides a standalone implementation, uniform differentiability across all tasks, native multi-agent support, and high-fidelity 3D environments.

## 3. FluidGym: Overview

Motivated by the limitations of existing AFC benchmarks, FluidGym is an ML-tailored framework designed around the following desiderata: (i) a standardized, standalone, and easy-to-use RL–CFD interface that runs entirely in Python without external CFD software, (ii) an end-to-end differentiable framework suitable for various control methodologies, (iii) inherent support of multi-agent control, and (iv) high-fidelity 3D tasks. In the following, we outline the core design principles underlying our benchmark and describe how FluidGym fulfills these desiderata.

### 3.1. Architecture and Interaction Interface

Figure 2 summarizes the architecture of FluidGym, which unifies CFD simulation and control under a single, RL-centric interface. To meet desiderata (i) and (ii), FluidGym integrates the GPU-accelerated PICT solver (Franz et al. (2026); see Appendix B) with a modular PyTorch (Ansel et al., 2024) interaction layer. Because the design runs entirely in PyTorch, environment stepping and backprop use the same autograd mechanisms as standard deep networks. Consequently, no external CFD software or coupling code

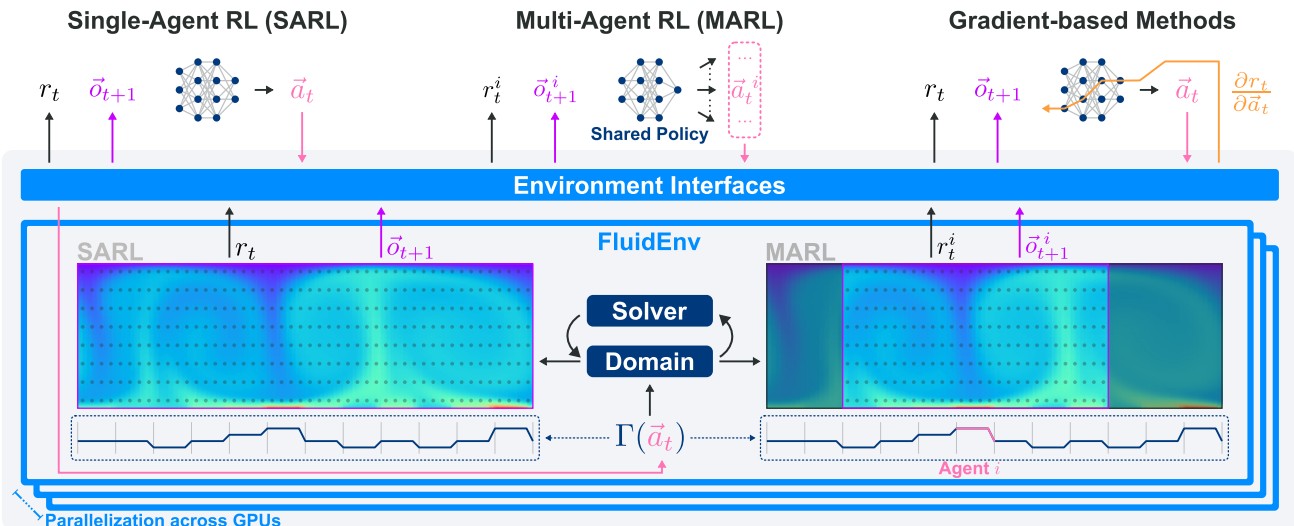

*Figure 2.* Overview of FluidGym using the 2D Rayleigh–Bénard Convection (RBC) environment. The framework provides three modes of interaction: single-agent RL (SARL), multi-agent RL (MARL), and gradient-based methods. The action space consists of 12 heater actuators along the lower boundary. In SARL, a single agent outputs the full action vector, whereas in MARL, each agent controls one actuator via a local action. Local actions are internally aggregated and mapped to boundary actuation values via the transformation function $\Gamma$. Gray dots indicate virtual sensor locations: in SARL, the agent receives all measurements, while in MARL, each agent observes only the local subset around its assigned actuator (denoted by the window framed in purple).

is required, and environments are compatible with common RL libraries through a lightweight API. The `FluidEnv` abstraction encapsulates all CFD computations and exposes standardized observation, action, and reward interfaces for both differentiable and classical RL methods. Finally, FluidGym scales to large experimental workloads via parallel execution of environments across multiple GPUs.

Addressing desideratum (iii), the `FluidEnv` is implemented from the ground up with both single-agent and multi-agent RL in mind. Its interface provides standardized observation, action, and reward specifications for centralized or decentralized control. All environments are modular, enabling new tasks to be defined by specifying domain configuration and control logic. This design makes FluidGym an extensible platform for future research on RL for AFC.

Finally, addressing desideratum (iv), our environments built on top of `FluidEnv` focus on state-of-the-art, high-fidelity 3D flow simulations (see Section 3.2).

**Modes of Interaction**  FluidGym supports three modes of interaction through its environment interfaces, which expose the `FluidEnv` via common RL environment interfaces, including Gymnasium (Towers et al., 2024), PettingZoo (Terry et al., 2021), Stable-Baselines3 (SB3, Raffin et al. (2021)), and TorchRL (Bou et al., 2023). First, in the single-agent RL (SARL) setting, a single RL agent applies a global action $\vec{a}_t$ at each control step $t$ and the environment returns a global observation $\vec{o}_{t+1}$ and scalar reward $r_t$. Secondly, in the multi-agent RL (MARL) configuration, multiple agents

act simultaneously at different spatial locations in the domain. Each agent selects a local action $\vec{a}_t^i$ and receives a local observation $\vec{o}_{t+1}^i$ and individual reward $\vec{r}_t^i$. Reward functions in these settings are typically constructed of a weighted sum of local and global properties of the domain. This interaction mode enables decentralized cooperation control strategies, where equivariance to translations allows us to deploy the same agent in all locations (Vasanth et al., 2024; Peitz et al., 2024). Lastly, in addition to standard RL, FluidGym supports gradient-based control methods by providing end-to-end differentiability with respect to the applied action. This allows gradients to be backpropagated through FluidGym to the policy parameters.

**Example: 2D Rayleigh–Bénard Convection**  Possible interaction modes are visualized in Figure 2 using the 2D Rayleigh-Bénard Convection (RBC) environment as an example. Here, the action space consists of 12 scalar control inputs corresponding to heater elements placed along the lower boundary of the domain. In the SARL scenario, a single agent outputs the complete action vector, assigning temperature intensities to all actuators. In contrast, in the MARL configuration, each agent controls one individual actor via its local action. Internally, the environment interface aggregates local actions into a global action vector. The resulting action vector is then transformed into physically meaningful boundary condition values via the control mapping function $\Gamma$, in this case, normalization and spatial smoothing. Observations are constructed from virtual sensor measurements represented by gray dots in the visualization.

*Table 2.* Overview of the FluidGym environments, listing control objectives, observation and action dimensions, SARL/MARL support, and mean per-step runtime across all difficulty levels on a single NVIDIA A100 GPU. SARL is omitted for environments with very large action spaces, where centralized control becomes impractical. For more details, see Table 4 in Appendix C and Table 8 in Appendix E.

| ID PREFIX | OBJECTIVE | #SENSORS | #ACTORS | SARL | MARL | RUNTIME [SEC/STEP] |
|---|---|---|---|---|---|---|
| CYLINDERROT2D | | 302 | 1 | ✓ | × | 1.95 |
| CYLINDERJET2D | DRAG REDUCTION | 302 | 1 | ✓ | × | 2.01 |
| CYLINDERJET3D | | 4832 | 8 | ✓ | ✓ | 9.52 |
| RBC2D | | 768 | 12 | ✓ | ✓ | 1.92 |
| RBC2D-WIDE | HEAT TRANSFER | 1 536 | 24 | ✓ | ✓ | 1.99 |
| RBC3D | ENHANCEMENT | 221 184 | 64 | × | ✓ | 1.17 |
| RBC3D-WIDE | | 884 736 | 256 | × | ✓ | 1.71 |
| AIRFOIL2D | AERODYNAMIC EFFICIENCY | 418 | 3 | ✓ | × | 28.76 |
| AIRFOIL3D | ENHANCEMENT | 2508 | 12 | ✓ | ✓ | 52.89 |
| TCFSMALL3D-BOTH | | 1 024 | 1 024 | × | ✓ | 0.33 |
| TCFSMALL3D-BOTTOM | DRAG REDUCTION | 512 | 512 | × | ✓ | 0.29 |
| TCFLARGE3D-BOTH | | 4 096 | 4 096 | × | ✓ | 0.56 |
| TCFLARGE3D-BOTTOM | | 2 048 | 2 048 | × | ✓ | 0.52 |

**Training and Evaluation Protocol** Many prior works lack standardized training and evaluation procedures for RL in AFC, with studies differing widely in how many and which initial conditions they use. FluidGym addresses this by providing a unified protocol based on three predefined splits (train, val, and test) each containing ten randomly generated initial domains. On first use, initial domains are automatically downloaded and cached locally. Each env.reset() applies random perturbations and random rollout steps; with consistent RNG seeding, this creates a standardized and reproducible train/val/test protocol.

## 3.2. Benchmark Environments

FluidGym provides a diverse set of environments, each introducing distinct challenges for learning well-performing RL policies. Formal SARL and MARL environment definitions are stated in Appendix A. Each environment is offered in three difficulty levels to introduce increasing levels of turbulence and flow complexity. An overview of the environments is shown in Figure 1 and summarized in Table 2. In the following, we outline four key flow scenarios, building the foundation of the 13 FluidGym environments.

**Flow Past a Cylinder** The *von Kármán vortex street* is a canonical setup in which flow separation behind a cylinder induces periodic vortex shedding and fluctuating forces on the cylinder (Schäfer et al., 1996). This configuration has consistently served as a benchmark for AFC using RL to reduce the drag acting on the cylinder (Koizumi et al., 2018; Rabault et al., 2019; Xu et al., 2020; Tang et al., 2020; Ren et al., 2021; Han et al., 2022; Suárez et al., 2025). The system is parametrized via the Reynolds number $\mathrm{Re} = \frac{\overline{U}D}{\nu}$ with mean incoming velocity $\overline{U}$, cylinder diameter $D$, and

kinematic viscosity $\nu$. The objective is to reduce the drag coefficient $C_D$ while keeping the lift $C_L$ small, using the reward $r_t = C_{D,\mathrm{ref}} - \langle C_D \rangle_{T_{\mathrm{act}}} - w|\langle C_L \rangle_{T_{\mathrm{act}}}|$, with lift regularization weight $w \geq 0$ and $\langle \cdot \rangle_{T_{\mathrm{act}}}$ referring to averaging over the actuation interval and reference uncontrolled drag coefficient $C_{D,\mathrm{ref}}$. We note that normalization with uncontrolled reference metrics is not essential in principle, but is used consistently across the benchmark. Actuation uses either (i) opposing synthetic jets on the top and bottom surfaces of the cylinder, or (ii) cylinder rotation. Difficulty levels, defined via $\mathrm{Re}$, span different flow regimes in 2D/3D.

**Rayleigh-Bénard Convection** The Rayleigh-Bénard Convection (RBC; Bénard (1900); Rayleigh (1916)) models a buoyancy-driven flow between a heated bottom plate and a cooled top plate. This leads to convective fluid motion and the formation of thermal plumes with complex, potentially chaotic patterns (Pandey et al., 2018). The system is defined by two dimensionless parameters, the Prandtl number $\mathrm{Pr}$ and the Rayleigh number $\mathrm{Ra}$. $\mathrm{Pr}$ is a material property of the fluid, while $\mathrm{Ra}$ controls the intensity of buoyancy-driven convection. Our setup follows Vignon et al. (2023), extended to 3D as in Vasanth et al. (2024), with the domain height reduced from 2 to 1 to match the standard dimensionless configuration (Pandey et al., 2018). The task aims to reduce convective heat transfer by minimizing the instantaneous Nusselt number $\mathrm{Nu}_{\mathrm{instant}} = \sqrt{\mathrm{RaPr}}\langle u_y T \rangle_V$, where $u_y$ denotes the vertical fluid velocity, $T$ the temperature field, and $\langle \cdot \rangle_V$ a volume average (Pandey et al., 2018), resulting in the reward $r_t = \mathrm{Nu}_{\mathrm{ref}} - \mathrm{Nu}_{\mathrm{instant}}$. Control is applied via bottom-boundary heaters whose temperatures are normalized, clipped, and spatially smoothed. The environment difficulty is varied by adjusting the Rayleigh number $\mathrm{Ra}$, with higher values in both 2D and 3D resulting

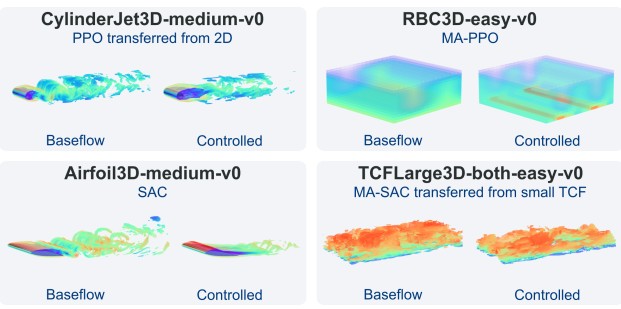

*Figure 3.* Final 3D flow fields at the end of test episodes for uncontrolled and controlled cases across four FluidGym environments using PPO, SAC, or multi-agent variants. Transfer cases use policies trained on corresponding 2D or smaller domains.

in more turbulent convection. An additional wide-domain variant with aspect ratio $2\pi$ introduces richer spatial patterns. As a reference, FluidGym provides a PD controller baseline (Beintema et al., 2020; Markmann et al., 2025).

**Flow Past an Airfoil** The flow around an airfoil is a fundamental configuration in aerodynamics and a common benchmark for AFC (Wang et al., 2022b; Garcia et al., 2025; Liu et al., 2025; Montalà et al., 2025). Variations in Reynolds number and angle of attack influence flow separation and vortex dynamics. Control aims to improve aerodynamic efficiency by increasing the lift to drag ratio, i.e., $r_t = \langle C_L \rangle_{T_{\text{act}}} / \langle C_D \rangle_{T_{\text{act}}} - C_{L,\text{ref}} / C_{D,\text{ref}}$. Actuation is provided by zero net-mass-flux synthetic jet actuators mounted on the airfoil surface. Task difficulty is set by the Reynolds number, with higher values producing sharper flow separation and stronger turbulence.

**Turbulent Channel Flow** The turbulent channel flow (TCF; the flow between two parallel, infinitely large plates) is a classic experiment for studying wall-bounded turbulence. Most AFC strategies aim to reduce the wall shear stress by imposing wall normal velocities (blowing or suction) via spatially distributed actuators at the walls (Bewley et al., 2001; Stroh et al., 2015; Guastoni et al., 2023; Sonoda et al., 2023; Zhao et al., 2025). The objective is captured through a reward based on the instantaneous reduction of shear stress $\tau_{\text{wall}}$ relative to the uncontrolled reference $\tau_{\text{wall,ref}}$, i.e., $r_t = 1 - \tau_{\text{wall}} / \tau_{\text{wall,ref}}$. FluidGym provides both a small and a large channel variant, enabling evaluation under different spatial scales. Additionally, FluidGym provides a pre-computed opposition control baseline for this environment consistent with previous work (Guastoni et al., 2023).

## 4. Experiments

In the following, we present a comprehensive evaluation of FluidGym. All experimental results and trained models are publicly available at `https://huggingface.co/datasets/safe-autonomous-systems/fluidgym-experiments`.

### 4.1. Experimental Setup

In our experiments, we evaluate Proximal Policy Optimization (PPO; Schulman et al. (2017)) and Soft Actor–Critic (SAC; Haarnoja et al. (2018)) using their Stable-Baselines3 (SB3; Raffin et al. (2021)) implementations, denoted as MA-PPO and MA-SAC in the MARL setting with a shared policy across actors. To enable the first large-scale evaluation of these algorithms on AFC, we use default SB3 hyperparameters (see Appendix D). Additionally, we provide a network size ablation for PPO in Appendix E.5 to ensure that our results reflect algorithmic performance rather than insufficient model capacity. To demonstrate the advantages of a fully differentiable benchmark for policy learning, we evaluate Differentiable Predictive Control (DPC; Drgoňa et al. (2022)) on the `CylinderJet2D` and `RBC2D` environments, alongside Temporal Difference Learning for Model Predictive Control (TD-MPC; Hansen et al. (2022)) as a representative model-based method for this subset.

We conduct all experiments using five random seeds and ten test episodes. We report per-step rather than cumulative metrics to avoid episode-length confounding; since environment episode lengths are constant, this preserves relative or normalized metrics. Hard 3D airfoil results are limited to MA-PPO, while medium tasks also include PPO and SAC.

### 4.2. Overall Benchmark Performance

Before presenting quantitative results, we first show exemplary final flow fields from controlled test set rollouts to illustrate the resulting flow states. Figure 3 displays four 3D environments with their uncontrolled and controlled cases at the end of test episodes, including transferred policies.

Then, to assess the overall performance of RL algorithms on FluidGym, we consider their respective performance profiles following Agarwal et al. (2021), which depict the tail distribution of normalized rewards aggregated across all environments and random seeds. Figure 4 (left) shows the profiles of PPO, SAC, and their respective multi-agent variants. Notably, the performance profiles of PPO and SAC vary substantially. We attribute this to a slower overall learning and convergence behavior (see Appendix E for detailed results). For the multi-agent variants, we observe similar performance profiles for both algorithms. MA-SAC exhibits marginally higher scores overall, though the differences partially lie within the corresponding confidence intervals.

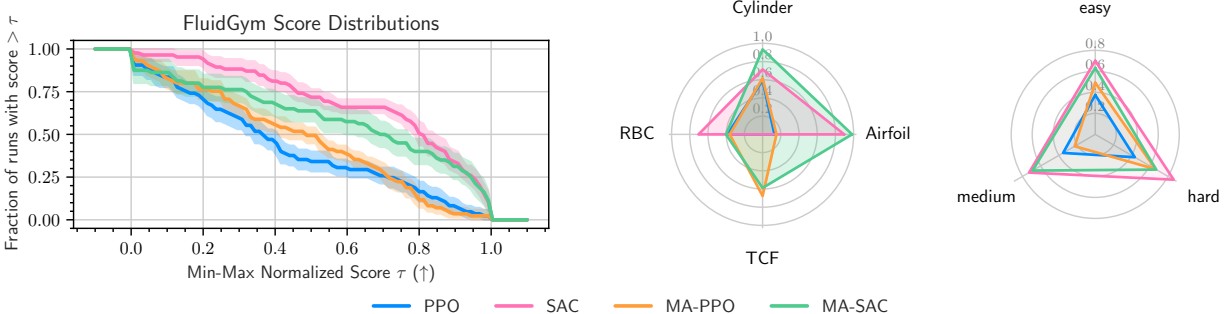

*Figure 4.* **Left:** Performance profiles as proposed by Agarwal et al. (2021) summarizing scores over all evaluated environments. Error bars indicate pointwise 95% confidence intervals based on 2 k stratified bootstrap replications across random seeds. **Right:** Interquartile mean (IQM) scores over environment classes (middle) and difficulty levels (right). For all panels, scores are computed as min–max normalized relative improvements over the baseflow, with normalization performed independently for each environment–difficulty pair.

Inspecting performance across environment categories and difficulty levels (Figure 4, right) shows a consistent pattern: SAC achieves the highest normalized test set relative improvement over the baseflow across all levels, while MA-PPO performs slightly better on the TCF environments.

Overall, two trends emerge: (i) SAC reliably outperforms PPO across all difficulty levels, while the multi-agent variants are more comparable, likely because PPO benefits from increased sample counts, which reduces SAC's usual sample-efficiency advantage; and (ii) environments with similar flow structures (e.g., cylinder and airfoil) yield similar learning dynamics and performance, despite differing reward definitions. These observations highlight the importance of algorithmic robustness and sample efficiency when scaling RL to turbulent AFC tasks.

### 4.3. Gradient-Based Learning of Control Policies

Besides standard RL, FluidGym enables policy learning via its end-to-end differentiability. DPC learns a policy *exclusively* via reward gradients propagated through the differentiable simulation, omitting value estimation, explicit exploration, or auxiliary components. As shown in Figure 5 (top), DPC achieves substantial speedups over RL on `CylinderJet2D-easy-v0`, outperforming PPO and SAC by approximately one and two orders of magnitude, respectively. In a more turbulent setting on `RBC2D-hard-v0`, DPC is competitive, performing on par with TD-MPC and SAC and clearly outperforming PPO.

Next, we discuss an individual test set episode using the final policies for `CylinderJet2D-easy-v0`. Figure 5 (bottom) shows the temporal evolution of applied actions $a_t$ and the resulting drag coefficients $C_D$. All three RL policies rapidly attenuate oscillations and reduce drag relative to the uncontrolled baseflow, with SAC achieving the lowest final drag coefficient corresponding to a drag reduction

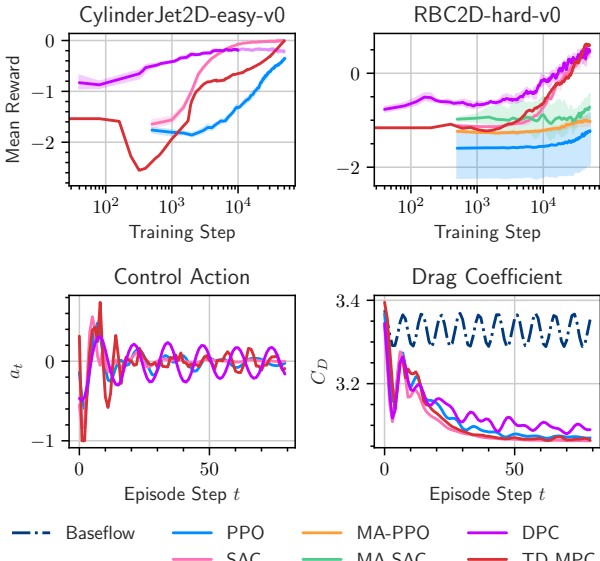

*Figure 5.* **Top:** Mean training reward over time on `CylinderJet2D-easy-v0` and `RBC2D-hard-v0`. Note: As DPC training degraded over time, the training was stopped after $10^4$ steps. **Bottom:** `CylinderJet2D-easy-v0`: Time evolution of the control action $a_t$ and drag coefficient $C_D$ for the uncontrolled baseflow, the final PPO, SAC, TD-MPC, and DPC policies on the `CylinderJet2D-easy-v0` test environment.

of approximately 8%, comparable to TD-MPC and PPO, and in agreement with findings by Rabault et al. (2019). The observed drag reduction of approximately 7.2% for the DPC policy indicates that reward gradients provide informative and stable control signals for AFC. This underscores the value of FluidGym as the first fully differentiable AFC benchmark and foundation for research on gradient-based learning of AFC strategies. DPC results are currently limited to the `CylinderJet2D` and `RBC2D` environments.

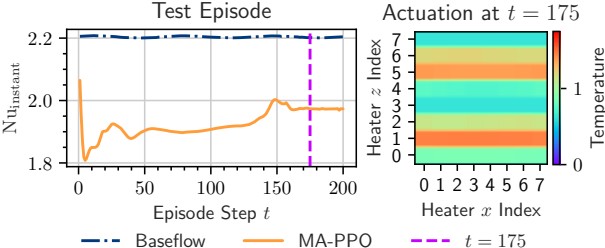

*Figure 6.* `RBC3D-easy-v0`: Time evolution of the Nusselt number $\mathrm{Nu_{instant}}$ for the baseflow and MA-PPO policy (left) and bottom-plate actuation at $T = 175$ (right) on the test environment.

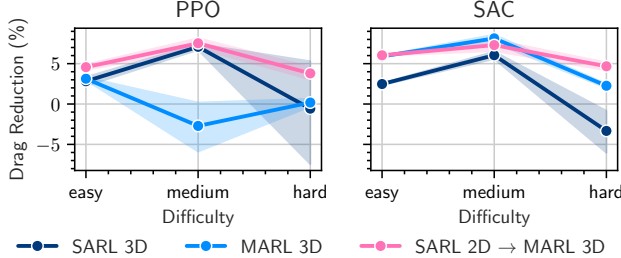

*Figure 7.* `CylinderJet3D`: Drag reduction across difficulty levels for PPO and SAC comparing SARL 3D, MARL 3D, and transferred SARL 2D → MARL 3D with 95% confidence intervals.

### 4.4. Emerging Multi-Agent Interaction

Beyond single-agent cylinder control, FluidGym also enables studying multi-agent AFC tasks. Figure 6 shows a test episode on `RBC3D-easy-v0` using MA-PPO, where agents coordinate bottom-wall heating to form two stable convection rolls. Notably, when investigating the actuation, we observe emerging coordinated behavior between the individual agents, leading to two separate convection rolls. These spatial heating patterns are consistent with the findings of Vasanth et al. (2024) and suggest that RL can learn a spatially invariant control policy forming globally coordinated behavior. This highlights the potential of MARL for AFC, a key capability of FluidGym.

### 4.5. Policy Transfer Across Environment Variations

We further evaluate policy transfer in FluidGym, considering (i) dimensionality transfer for the cylinder flow and (ii) domain-size transfer for the TCF.

**Transfer across Dimensionalities** We investigate how policies trained in 2D transfer to their 3D counterparts using the cylinder environment. Figure 7 shows the mean test set drag reduction for three approaches: 3D SARL and MARL trained in 3D, and a transferred 2D → 3D policy applied to the eight actuators in 3D individually. On the easy task,

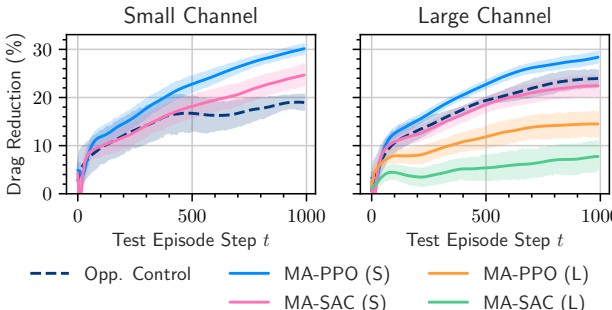

*Figure 8.* TCF: Mean test-episode drag reduction with 95% confidence intervals for opposition control (Opp. Control) as well as policies trained on the small (S) and large (L) channel, respectively.

the transferred policy outperforms the 3D-trained baselines. On medium difficulty, it is on par, slightly below PPO and MA-SAC. On the hard task, it again achieves the highest drag reduction. These findings indicate that direct transfer from 2D to 3D can be robust despite the added complexity.

**Transfer across Domain Sizes** Finally, we study whether policies trained in smaller TCF domains transfer to larger ones. This setting is motivated by two factors: (i) lower simulation cost in smaller domains, and (ii) MARL may yield control policies that are translation-equivariant and thus insensitive to the absolute domain size. Figure 8 shows mean test-episode drag reduction in the large domain for policies trained either on the small channel (S) or directly on the large channel (L), together with an opposition control baseline. Notably, policies trained in the small domain perform comparably to opposition control and substantially outperform those trained directly in the large domain. This demonstrates that MARL can learn spatially transferable control strategies, enabling policy learning in simplified domains and generalization to more complex flow settings.

### 4.6. Algorithm Runtime Comparisons

Figure 9 summarizes algorithmic wall-clock times across FluidGym environment sets. Across all environment categories, three patterns can be observed: (i) PPO and SAC exhibit similar training times except for the TCF environments, most likely due to high transition counts in the massively parallel setting; (ii) TD-MPC exhibits the lowest training runtimes among standard RL algorithms; and (iii) DPC leads to $1.5 - 2\times$ higher training times compared to standard RL, due to the extra backward pass through the differentiable environment. This highlights the inherent trade-off between the high sample efficiency of DPC and the increased wall-clock time required for its gradient-based policy updates.

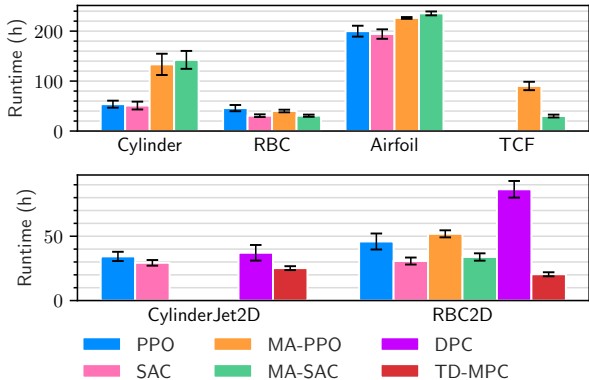

*Figure 9.* Training wall-clock times of algorithms for different sets of environments. **Top:** Average training time per algorithm over all FluidGym environment classes. **Bottom:** Average training time per algorithm over specific environments with DPC and TD-MPC.

## 5. Limitations and Future Work

While FluidGym provides a unified and extensible platform for studying RL for AFC, limitations remain. First, the current evaluation is based on a limited number of random seeds due to the substantial computational cost associated with CFD simulations. As a result, the statistical robustness of the results is still limited when it comes to comparisons between algorithms. Second, FluidGym currently requires a CUDA-enabled GPU for fast simulation, as the underlying solver depends on custom CUDA kernels. Although installation is simplified through pre-built wheels, CPU-only execution is not yet supported. Third, despite full differentiability of FluidGym, we focus on RL and only demonstrate DPC leveraging reward gradients on a subset of environments. Systematic comparisons with other differentiable control approaches are not included. Finally, baseline algorithms are evaluated using standard hyperparameters from off-the-shelf libraries, which promotes comparability but may not reflect each algorithm's optimal performance. Overall, these limitations stem from computational and practical considerations rather than inherent constraints of FluidGym.

Several directions offer potential for extending FluidGym and broadening its utility and scope. First, increasing the number of random seeds used during training and evaluation will improve the statistical robustness of the reported baseline results. Expanding the DPC experiments to more complex 3D environments is a promising future direction for gradient-based learning. Additionally, evaluating differentiable RL (Xu et al., 2022; Xing et al., 2025), that combines gradient-based control with classical RL, is a natural next step. Expanding the set of environments to cover additional geometries and physical regimes would provide a more comprehensive assessment of control strategies across diverse flow configurations. Beyond incompress-

ible Navier–Stokes, we also plan to extend FluidGym to magnetohydrodynamic (MHD) flows, enabling the study of control in electrically conducting fluids (e.g., in fusion-relevant settings). Finally, we intend to add progressively more challenging environments as control methods advance to keep the benchmark aligned with the state of the art.

## 6. Conclusion

In this work, we introduce FluidGym, the first standalone, fully differentiable benchmark suite for reinforcement learning in active flow control. By combining a GPU-accelerated CFD solver with a standardized RL interface, FluidGym removes the dependency on external CFD code and provides a unified, accessible, and reproducible platform that bridges RL research and fluid dynamics. Our benchmark suite provides diverse 2D and 3D environments with consistent observation, actuation, and reward definitions, unified evaluation protocols, and support for single- and multi-agent RL as well as gradient-based methods.

PPO and SAC baselines align with prior findings and show FluidGym's suitability for RL and gradient-based control, with DPC demonstrating the effectiveness of leveraging reward gradients for AFC. By releasing all environments and trained models, we aim to lower the barrier to entry for researchers and foster reproducibility and comparability.

## Acknowledgements

JB and SP acknowledge funding from the European Research Council (ERC Starting Grant "KoOpeRaDE") under the European Union's Horizon 2020 research and innovation programme (Grant agreement No. 101161457). The computations were performed on the compute cluster of the Lamarr Institute for Machine Learning and Artificial Intelligence, as well as on the high-performance computer "Noctua 2" at the NHR Center Paderborn Center for Parallel Computing (PC2), both of which are funded by the Federal Ministry of Research, Technology and Space and by the state of Northrhine-Westfalia.

## Impact Statement

In this work, we introduce a benchmark suite for reinforcement learning in active flow control with the goal of improving algorithms and policies for controlling fluid systems. Potential positive societal impacts include more energy-efficient transport and industrial processes, emission reduction, energy harvesting, and improved study of fluid flows.

At the same time, deploying learning-based controllers in safety-critical settings without rigorous validation could pose risks. The environments in FluidGym are idealized and do not capture the full complexity, including uncertainties

and constraints of real systems. Training reinforcement learning algorithms on high-fidelity simulations can also be computationally expensive and energy-consuming, which motivates future work on more sample-efficient algorithms.

Overall, this benchmark is a research tool to advance control methods for fluid systems, and we do not foresee direct societal harms associated with its use.

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

## A. Reinforcement Learning for Active Flow Control

Based on the definition by Sutton & Barto (1998), a reinforcement learning (RL) agent interacts with a Markov decision process (MDP) $\mathcal{M} = (\mathcal{S}, \mathcal{A}, R, T)$ with finite set of states $\mathcal{S}$, finite set of actions $\mathcal{A}$, reward function $R : \mathcal{S} \times \mathcal{A} \mapsto \mathbb{R}$, and transition function $T : \mathcal{S} \times \mathcal{A} \mapsto \mathcal{S}$. We note that we focus on deterministic MDPs here and do consider the discount factor $\gamma$ as RL hyperparameter and not as part of the MDP.

At each time step $t$, the agent selects an action $a_t = \pi(s_t)$ based on its policy $\pi$. In practice, $\pi$ is often described by a neural network with parameters $\theta$ and therefore denoted as $\pi_\theta$. Then, the environment returns the next state $s_{t+1}$ and a reward $r_t$ computed by $R(s_t, a_t)$.

When RL is applied to active flow control (AFC), the information based on which the agent selects its action is typically not the full state $s_t$ but a set of sensor observations $o_t$. This can be formalized as a partially observable Markov decision process (POMDP, Kaelbling et al. (1998)), which extends the MDP tuple by a finite set of observations $\Omega$ and observation function $O : \mathcal{S} \times \mathcal{A} \mapsto \Omega$. Again, we only consider POMDPs here. This leads to the following MDP definition for single-agent RL (SARL) used in this paper: $\mathcal{M}_{\text{SARL}} = (\mathcal{S}, \mathcal{A}, R, T, \Omega, O)$.

Based on the definition of a partially observable stochastic game (POSG, Albrecht et al. (2024)), we can extend our SARL definition to multiple agents. However, in the following, we again consider deterministic scenarios. In this setting, we consider $i \in I$ individual agents. While the sets of states, actions, and observations are shared between agents, each agent $i$ has an individual observation function $O_i$ and reward function $R_i$. This leaves us with the following MDP: $\mathcal{M}_{\text{MARL}} = (I, \mathcal{S}, \mathcal{A}, R_i, T, \Omega, O_i)$.

## B. The PICT Solver

In the following, we describe the core numerical details of the PICT solver (Franz et al., 2026) and provide numerical evidence to validate the underlying simulation of our benchmark.

### B.1. The PISO Algorithm

The Pressure Implicit with Splitting of Operators (PISO) algorithm introduced by Issa (1986) is a common method for the simulation of incompressible flows, which are governed by the Navier-Stokes equations (Navier, 1827; Stokes, 1845), consisting of the momentum equation

$$\frac{\partial \mathbf{u}}{\partial t} + \nabla \cdot (\mathbf{u}\mathbf{u}) - \nu \nabla^2 \mathbf{u} = -\nabla p + \mathbf{S} \tag{1}$$

and the continuity equation

$$\nabla \cdot \mathbf{u} = 0, \tag{2}$$

with time $t$, velocity $\mathbf{u}$, pressure $p$, viscosity $\nu$, and external source term $S$.

The PISO algorithm consists of two main procedures: (i) A predictor step, which advances the simulation and produces a predicted velocity $\mathbf{u}^*$, and (ii) typically two predictor steps computing the pressure, which is then used to make the predicted velocity $\mathbf{u}^*$ divergence free. In PICT, the PISO algorithm is discretized using the finite volume method (FVM; Kajishima & Taira (2017); Maliska (2023)) on a collocated grid. For the time advancement, the implicit Euler scheme is used. For buoyancy-driven convection, we employ the Boussinesq approximation.

### B.2. Gradient Computation

Simulation gradients in PICT are obtained via a combination of the Discretize-then-Optimize (DtO) and Optimize-then-Discretize (OtD) paradigms, with DtO applied to the global algorithmic structure and OtD to the inner linear system solves.

### B.3. Validation

First and foremost, the PICT solver was numerically validated by Franz et al. (2026). Additionally, we provide numerical evidence for the correctness of the environments in FluidGym.

**Flow Past a Cylinder**    For the cylinder, the temporal mean of the uncontrolled drag coefficient of 3.328 closely aligns with the value of approximately 3.205 reported by Rabault et al. (2019), resulting in a relative deviation of 3.84%. We partially attribute this to the difference between a non-reflecting advective outflow boundary in PICT and the free-stress boundary condition implemented by Rabault et al. (2019). Nevertheless, as described in Section 4.3, the resulting drag reductions achieved by the RL policies match both quantitatively and qualitatively.

*Table 3.* RBC grid refinement study. Reported Nusselt numbers correspond to the temporal mean of $\mathrm{Nu_{instant}}$ over 10 uncontrolled episodes.

| x **RESOLUTION** | $\langle\mathrm{Nu_{instant}}\rangle$ | **#CELLS** |
|---|---|---|
| 96 | 4.896 | 5 856 |
| 144 | 4.755 | 13 248 |
| 192 | 4.786 | 23 242 |

**Rayleigh-Bénard Convection**    Prior work has largely relied on numerical setups that differ from the standard non-dimensional formulation (Pandey et al., 2018), partially yielding inconsistent Nusselt numbers (Vignon et al., 2023; Markmann et al., 2025). To validate our environment, we perform a grid refinement study (Table 3). The grid with resolution 96 shows a relative deviation of 2.298%, and demonstrates learning behavior consistent with previous studies (Vignon et al., 2023).

**Flow Past an Airfoil**    For the airfoil, we obtain a mean drag coefficient of 0.278 and a mean lift coefficient of 0.993. These values compare well with those reported by Wang et al. (2022b), who obtained an average drag of 0.324 and an average lift of 1.003. We note that our computational domain is longer (6 chord lengths versus 3.5), which accounts for part of the discrepancy. Nevertheless, we observe consistent quantitative and qualitative behavior across all flow states.

**Turbulent Channel Flow**    For the channel configuration, we adopt the same numerical setup previously validated by Franz et al. (2026), including the wall-stress computation used in the forcing term. Therefore, additional baseline validation is not required. Our opposition-control case yields a 20% drag reduction, and the RL-controlled case reaches 30%, both of which are in close agreement with prior work (Guastoni et al., 2023).

# C. Environments

*Table 4.* Difficulty levels and corresponding physical parameters for all FluidGym environments. Cylinder and airfoil tasks are parameterized by the Reynolds number $\text{Re}$, RBC by the Rayleigh number $\text{Ra}$, and turbulent channel flow (TCF) by the friction Reynolds number $\text{Re}_\tau$. Actions are applied over a period of $T_{\text{act}}$ with numerical time step $\Delta t$. **Note:** For the TCF environments, $T_{\text{act}}$ and $\Delta t$ in the table refer to $T_{\text{act},+}$ and $\Delta t_+$, respectively.

| ID PREFIX | DIFFICULTY | PARAMETER | VALUE | DOMAIN SIZE ($L \times H[\times D]$) | $T_{\text{act}}$ | $\Delta t$ |
|---|---|---|---|---|---|---|
| `CylinderRot2D` | EASY | Re | 100 | $22 \times 4.1$ | 0.25 | 0.01 |
| | MEDIUM | Re | 250 | $22 \times 4.1$ | 0.25 | 0.01 |
| | HARD | Re | 500 | $22 \times 4.1$ | 0.25 | 0.01 |
| `CylinderJet2D` | EASY | Re | 100 | $22 \times 4.1$ | 0.25 | 0.01 |
| | MEDIUM | Re | 250 | $22 \times 4.1$ | 0.25 | 0.01 |
| | HARD | Re | 500 | $22 \times 4.1$ | 0.25 | 0.01 |
| `CylinderJet3D` | EASY | Re | 100 | $22 \times 4.1 \times 4$ | 0.25 | 0.01 |
| | MEDIUM | Re | 250 | $22 \times 4.1 \times 4$ | 0.25 | 0.01 |
| | HARD | Re | 500 | $22 \times 4.1 \times 4$ | 0.25 | 0.01 |
| `RBC2D` | EASY | Ra | $8 \times 10^4$ | $\pi \times 1$ | 1.0 | 0.05 |
| | MEDIUM | Ra | $4 \times 10^5$ | $\pi \times 1$ | 1.0 | 0.05 |
| | HARD | Ra | $8 \times 10^5$ | $\pi \times 1$ | 1.0 | 0.05 |
| `RBC2D-wide` | EASY | Ra | $8 \times 10^4$ | $2\pi \times 1$ | 1.0 | 0.05 |
| | MEDIUM | Ra | $4 \times 10^5$ | $2\pi \times 1$ | 1.0 | 0.05 |
| | HARD | Ra | $8 \times 10^5$ | $2\pi \times 1$ | 1.0 | 0.05 |
| `RBC3D` | EASY | Ra | $6 \times 10^3$ | $\pi \times 1 \times \pi$ | 1.0 | 0.05 |
| | MEDIUM | Ra | $8 \times 10^3$ | $\pi \times 1 \times \pi$ | 1.0 | 0.05 |
| | HARD | Ra | $1 \times 10^4$ | $\pi \times 1 \times \pi$ | 1.0 | 0.05 |
| `RBC3D-wide` | EASY | Ra | $6 \times 10^3$ | $2\pi \times 1 \times 2\pi$ | 1.0 | 0.05 |
| | MEDIUM | Ra | $8 \times 10^3$ | $2\pi \times 1 \times 2\pi$ | 1.0 | 0.05 |
| | HARD | Ra | $1 \times 10^4$ | $2\pi \times 1 \times 2\pi$ | 1.0 | 0.05 |
| `Airfoil2D` | EASY | Re | $1 \times 10^3$ | $6 \times 1.4$ | 0.25 | 0.05 |
| | MEDIUM | Re | $3 \times 10^3$ | $6 \times 1.4$ | 0.25 | 0.05 |
| | HARD | Re | $5 \times 10^3$ | $6 \times 1.4$ | 0.25 | 0.05 |
| `Airfoil3D` | EASY | Re | $1 \times 10^3$ | $6 \times 1.4 \times 1.4$ | 0.25 | 0.05 |
| | MEDIUM | Re | $3 \times 10^3$ | $6 \times 1.4 \times 1.4$ | 0.25 | 0.05 |
| | HARD | Re | $5 \times 10^3$ | $6 \times 1.4 \times 1.4$ | 0.25 | 0.05 |
| `TCFSmall3D-both` | EASY | $\text{Re}_\tau$ | 180 | $\pi \times 2 \times \pi/2$ | 0.6 | 0.06 |
| | MEDIUM | $\text{Re}_\tau$ | 330 | $\pi \times 2 \times \pi/2$ | 0.6 | 0.06 |
| | HARD | $\text{Re}_\tau$ | 550 | $\pi \times 2 \times \pi/2$ | 0.6 | 0.06 |
| `TCFSmall3D-bottom` | EASY | $\text{Re}_\tau$ | 180 | $\pi \times 2 \times \pi/2$ | 0.6 | 0.06 |
| | MEDIUM | $\text{Re}_\tau$ | 330 | $\pi \times 2 \times \pi/2$ | 0.6 | 0.06 |
| | HARD | $\text{Re}_\tau$ | 550 | $\pi \times 2 \times \pi/2$ | 0.6 | 0.06 |
| `TCFLarge3D-both` | EASY | $\text{Re}_\tau$ | 180 | $2\pi \times 2 \times \pi$ | 0.6 | 0.06 |
| | MEDIUM | $\text{Re}_\tau$ | 330 | $2\pi \times 2 \times \pi$ | 0.6 | 0.06 |
| | HARD | $\text{Re}_\tau$ | 550 | $2\pi \times 2 \times \pi$ | 0.6 | 0.06 |
| `TCFLarge3D-bottom` | EASY | $\text{Re}_\tau$ | 180 | $2\pi \times 2 \times \pi$ | 0.6 | 0.06 |
| | MEDIUM | $\text{Re}_\tau$ | 330 | $2\pi \times 2 \times \pi$ | 0.6 | 0.06 |
| | HARD | $\text{Re}_\tau$ | 550 | $2\pi \times 2 \times \pi$ | 0.6 | 0.06 |

Initial domains are publicly available in our HuggingFace dataset at https://huggingface.co/datasets/safe-autonomous-systems/fluidgym-data. All environments provide a unified action space of $[-1, 1]$ and scale the actions internally. A summary of all environments is stated in Table 4. We note that the medium and hard cases for the 3D Airfoil environment are not considered in this work due to computational limitations.

### C.1. Numerical Setup

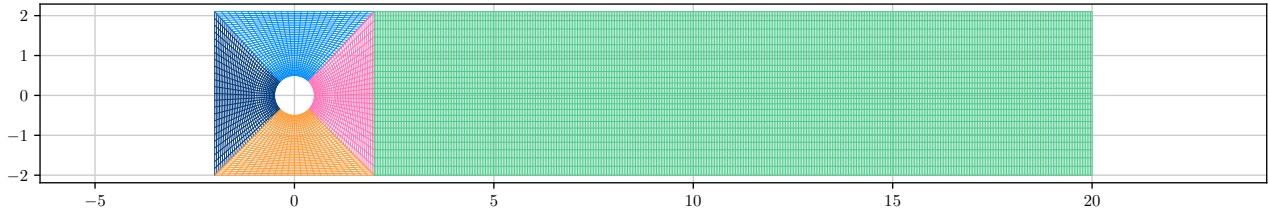

*Figure 10.* Computational domain of the 2D cylinder environment with easy difficulty. The number of angular cells $N_{\mathrm{angular}}$ is defined and based on it, the remaining resolution specifications are computed dynamically. For higher difficulty levels, the number of cells is increased. For 3D, the domain is extruded spanwise with $N_{\mathrm{angular}}$ cells.

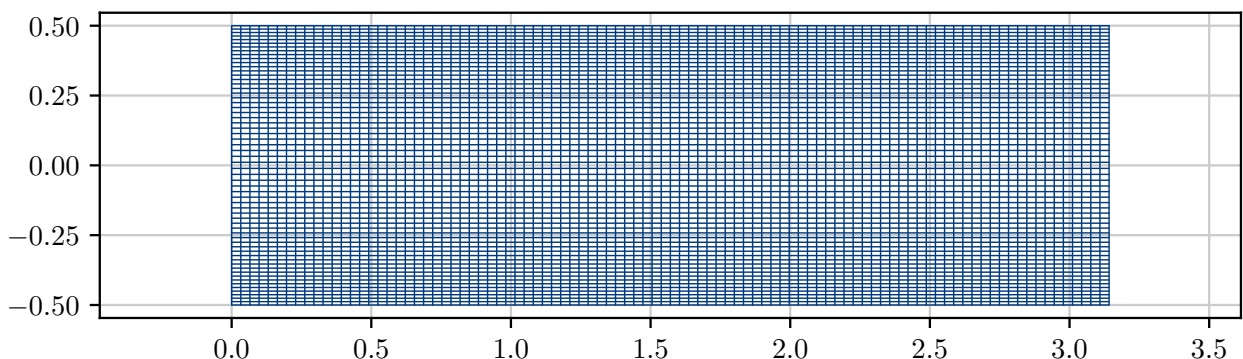

*Figure 11.* Computational domain of the 2D RBC environments, with a total resolution of $96 \times 61$ cells. The grid remains constant across difficulty levels and is refined towards the walls exponentially with a base of $1.02$. For 3D, the grid is expanded spanwise with 96 cells.

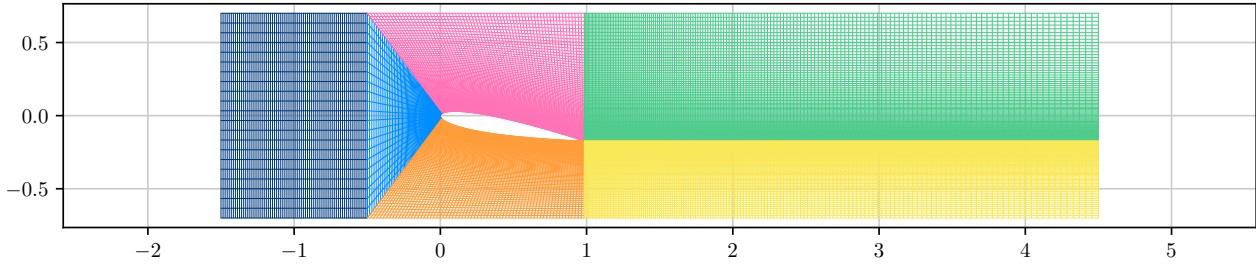

*Figure 12.* Computational domain of the airfoil environments. The grid is shared across all difficulty levels and extruded spanwise with 96 grid cells for 3D.

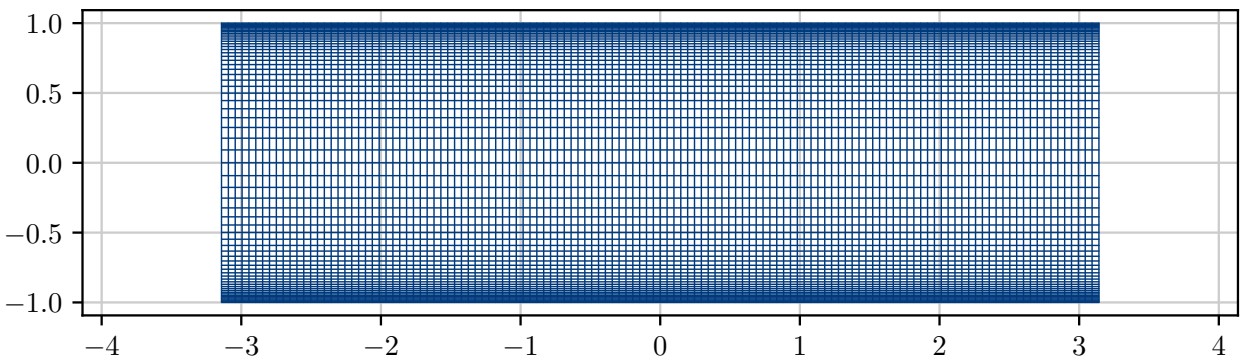

*Figure 13.* Computational domain of the large 3D TCF environments, with a total resolution of $128 \times 65 \times 128$ cells. The grid is refined towards the walls with a refinement strength of 2.

Domain sizes, actuation periods, and numerical time steps are stated in Table 4. All environments use adaptive time-stepping with a CFL-condition of $0.8$. For the TCF environments, the CFL-condition is set to $0.1$ following the numerical setup of Franz et al. (2026).

### C.2. Flow Past Cylinder

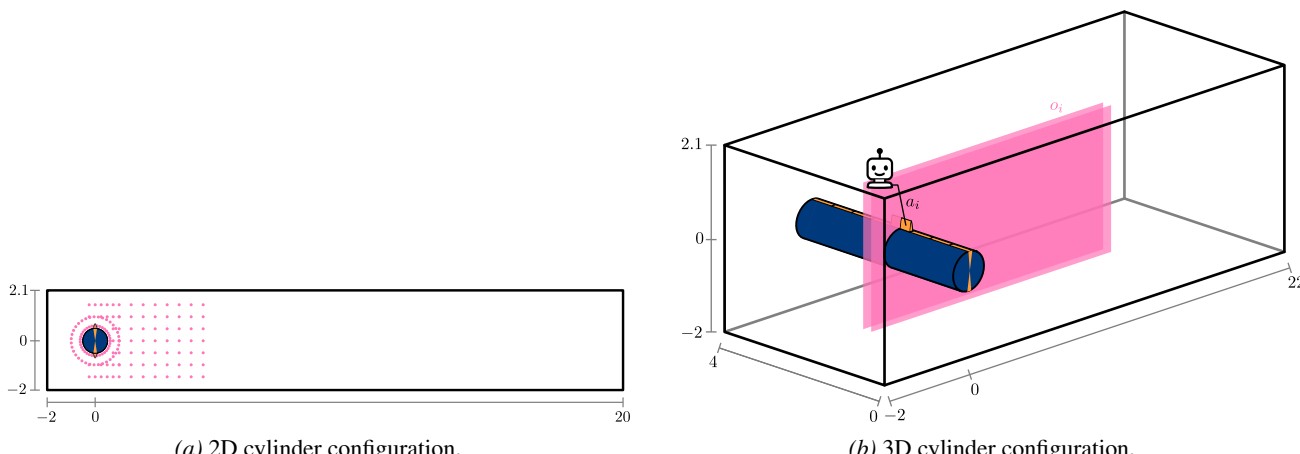

*(a)* 2D cylinder configuration.          *(b)* 3D cylinder configuration.

*Figure 14.* Overview of the 2D and 3D cylinder environments used in our benchmark. Jets are shown in orange and feature a parabolic profile with a total deflection angle of $10°$. Sensor locations are indicated in pink (dots in 2D, planes in 3D, with sensors placed analogously within each plane). In 3D, the domain is extended along the spanwise direction, yielding eight individual jet pairs.

**Reward Function**    The objective is to reduce the drag coefficient $C_D$ of the cylinder. Thus, the reward at step $t$ is defined as $r_t = C_{D,\text{ref}} - \langle C_D \rangle_{T_{\text{act}}} - \omega |\langle C_L \rangle_{T_{\text{act}}}|$, where the lift penalty $\omega$ is set to 1.0 as proposed by Ren et al. (2021) and the reference value corresponds to the uncontrolled baseline. $\langle \cdot \rangle_{T_{\text{Aact}}}$ corresponds to the temporal average over an actuation period, i.e., the simulation steps where the agent's actions are kept fixed. Following Rabault et al. (2019), the respective drag and lift coefficients are computed as

$$C_D = \frac{F_D}{\frac{1}{2}\rho \overline{U}^2 D} \text{ and } C_L = \frac{F_L}{\frac{1}{2}\rho \overline{U}^2 D} \tag{3}$$

$$\tag{4}$$

with the density $\rho = 1$ and forces acting on the cylinder

$$F_D = \int_S (\boldsymbol{\sigma} \cdot \mathbf{n}) \cdot \mathbf{e}_x \, \mathrm{d}S \text{ and } F_L = \int_S (\boldsymbol{\sigma} \cdot \mathbf{n}) \cdot \mathbf{e}_y \, \mathrm{d}S. \tag{5}$$

Here, $\boldsymbol{\sigma}$ is the Cauchy stress tensor, $\boldsymbol{n}$ the unit normal vector at the cylinder surface $S$ pointing into the fluid, and $\boldsymbol{e}_x = (1, 0, 0)$ and $\boldsymbol{e}_y = (0, 1, 0)$ the normal vectors along the $x$ and $y$ directions, respectively. In the MARL case, individual agent rewards are computed as $r_t^i = \beta\, r_t^{i,\text{local}} + (1 - \beta)\, r_t^{\text{global}}$. Local rewards are computed over the cylinder segment controlled by agent $i$, whereas the global reward is computed for the full cylinder. The local reward weight $\beta$ defines the impact of the local rewards and is set to $0.8$ following Suárez et al. (2025).

**Actuation**  Our 2D setups are based on jet actuators with a parabolic profile (Rabault et al., 2019) and cylinder rotation (Tokarev et al., 2020) with a maximum absolute value of $\overline{U}$ for the jet and rotation velocity, respectively. We further extend the jet actuation setup to 3D, following a setup similar to previous work (Suárez et al., 2025). Additionally, as proposed by Rabault et al. (2019), the action is smoothed over time using $c_s = c_{s-1} + \alpha(a_t - c_{s-1})$, where $c_s$ denotes the applied control value at simulation sub-step $s$ given the current action $a_t$ at episode step $t$ and previous control step $c_{s-1}$, and smoothing parameter $\alpha$ set to 0.1.

**Observations**  Observations consist of vertical and horizontal velocity components at the sensor locations indicated in Figure 14. In 3D, the observations also include the spanwise velocity component. To enable transfer from 2D to 3D, the number of sensor planes as well as the included velocity components can be set to match the 2D observations.

**Difficulty Levels**  Difficulty is defined via the Reynolds number (Re) and we use `easy` at $\mathrm{Re} = 100$, `medium` at $\mathrm{Re} = 250$, and `hard` at $\mathrm{Re} = 500$. Higher Reynolds numbers increase turbulence intensity and flow unsteadiness, which makes control more challenging. The `medium` and `hard` settings introduce three-dimensional flow interactions (Williamson, 1996).

### C.3. Rayleigh-Bénard Convection

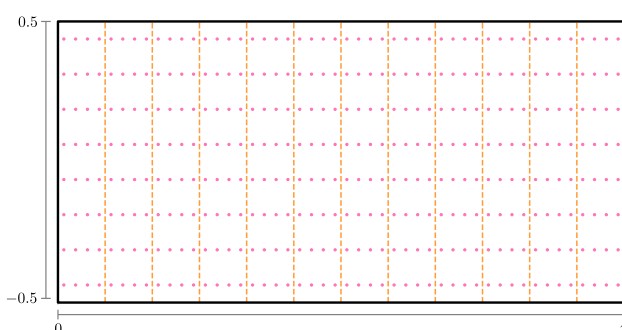

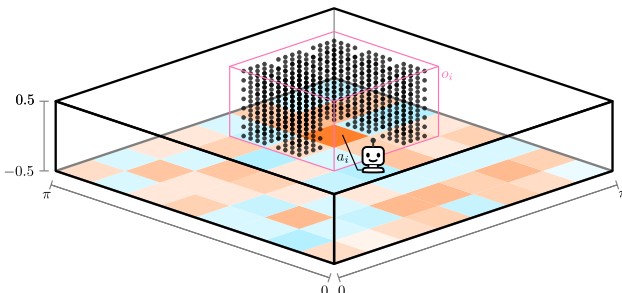

*(a)* 2D RBC configuration. Dashed lines indicate heater segments used for actuation.

*(b)* 3D RBC configuration. Actuation is applied via discretized heater patches along the bottom boundary. Each agent receives temperature and velocity observations within a local window of size 3 surrounding its actuator.

*Figure 15.* Overview of the 2D and 3D Rayleigh–Bénard convection (RBC) environments used in our benchmark. Control is provided through thermal actuation applied at the bottom boundary, while the top boundary is held at a fixed lower temperature. The environments support both centralized and decentralized control depending on the number and placement of actuators. For 3D, we omit centralized control in this work due to the large number of actuators.

**Reward Function**  The objective is to reduce convective heat transfer. We use the instantaneous dimensionless Nusselt number $\mathrm{Nu}_{\text{instant}} = \sqrt{\mathrm{RaPr}}\langle u_y T\rangle_V$ (Pandey et al., 2018) as performance measure, where $\langle \cdot \rangle_V$ denotes spatial averaging over the domain. The reward is defined as $r_t = \mathrm{Nu}_{\text{ref}} - \mathrm{Nu}_{\text{instant}}$, where the reference Nusselt number corresponds to the uncontrolled case.

**Actuation**   The control is implemented via localized heaters at the bottom boundary. Before being applied to the domain, the heater temperatures are normalized and clipped to ensure a mean of the default bottom temperature and a maximum heater temperature of $1.75$. Additionally, spatial smoothing is applied to avoid hard transitions in temperature between neighboring heaters, following Vignon et al. (2023) for 2D and Vasanth et al. (2024) for 3D.

**Observations**   Observations include all velocity components and the temperature at the sensor locations shown in Figure 15. For 2D, the default window size contains sensors above a total of 11 heaters, centered around the current actuated heater. For 3D, a window of $3 \times 3$ heaters and their associated sensors is included in the observations.

**Difficulty Levels**   We vary the Rayleigh number (Ra) to adjust the turbulence intensity. In 2D: `easy` at $\mathrm{Ra} = 8 \cdot 10^4$ (Vignon et al., 2023), `medium` at $\mathrm{Ra} = 4 \cdot 10^5$, and `hard` at $\mathrm{Ra} = 8 \cdot 10^5$. In 3D: `easy` at $\mathrm{Ra} = 6 \cdot 10^3$ (Vasanth et al., 2024), `medium` at $\mathrm{Ra} = 8 \cdot 10^3$, and `hard` at $\mathrm{Ra} = 10^4$. Higher Rayleigh numbers lead to stronger plume interactions and increasingly chaotic convection patterns.

**PD-Controller Baseline**   To provide a reference point for task difficulty, we include a linear proportional-derivative (PD) controller as proposed by previous works (Remillieux et al., 2007; Beintema et al., 2020; Markmann et al., 2025). The controller computes the bottom temperature actuation as

$$a(x,t) = k_p E(x,t) + k_d \frac{\Delta E(x,t)}{\Delta t},$$

where

$$E(x,t) = \langle u_y(x,y,t) \rangle,$$

and $k_p = 970$ and $k_d = 2000$ (Markmann et al., 2025).

### C.4. Flow Past Airfoil

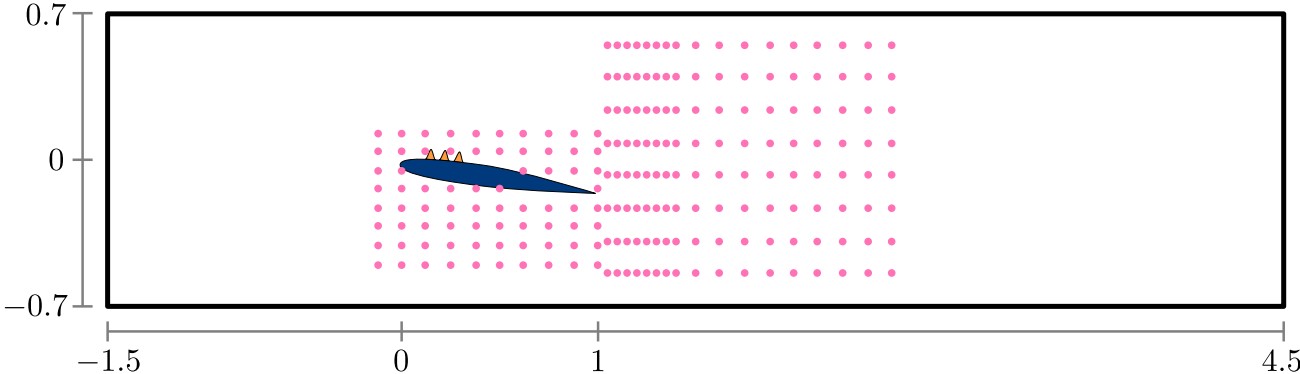

*Figure 16.* Schematic visualization of the 2D airfoil control environment. A stationary NACA 0012 airfoil is immersed in a uniform inflow at an angle of attack of $20°$. Actuation is provided through surface-mounted blowing and suction jets distributed along the airfoil surface (highlighted in orange), and sensors are placed at the pink marker locations. The corresponding 3D configuration follows the same setup but extends the domain spanwise with a depth of $D = 1.4$. In 3D, the actuation is discretized into four spanwise jet segments, yielding 12 individual actuators. In the MARL setting, each agent controls a group of three adjacent jets (one spanwise segment), enabling decentralized control.

**Reward Function**   The objective is to improve aerodynamic efficiency by increasing lift relative to drag. The reward at timestep is defined as

$$r_t = \frac{\langle C_L \rangle_{T_{\mathrm{act}}}}{\langle C_D \rangle_{T_{\mathrm{act}}}} - \frac{C_{L,\mathrm{ref}}}{C_{D,\mathrm{ref}}}, \qquad (6)$$

where $C_L$ and $C_D$ denote lift and drag coefficients, respectively, and averaging is performed over the actuation interval $T_{\mathrm{act}}$. The reference value corresponds to the uncontrolled baseline.

**Actuation**    Actuation is implemented using surface-mounted synthetic jet actuators placed on top of the airfoil (Garcia et al., 2025). A zero-net mass flux is enforced. As in the cylinder environment, the raw RL control signal is temporally filtered using exponential smoothing to ensure physically consistent actuation with $\alpha = 0.1$.

**Observations**    Observations follow the definition for the cylinder flow with sensor locations as shown in Figure 16. As the 3D case is extended similarly to the cylinder, we only visualize the 2D case.

**Difficulty Levels**    Difficulty is determined by the Reynolds number (Re), where higher Reynolds numbers lead to more abrupt separation and larger turbulence, which increases the challenge of effective flow control. We define `easy` at $Re = 10^3$, `medium` at $Re = 3 \cdot 10^3$, and `hard` at $Re = 5 \cdot 10^3$.

### C.5. Turbulent Channel Flow

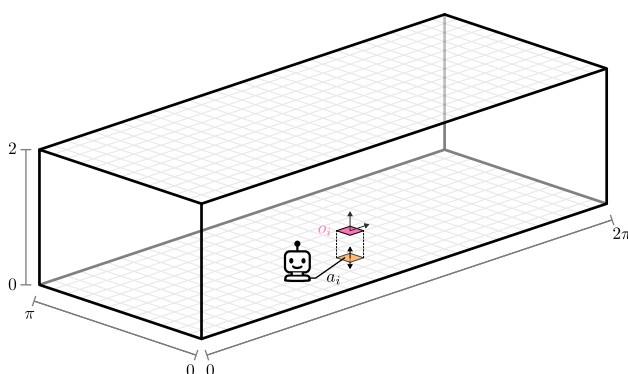

*Figure 17.* Schematic visualization of the large TCF environment. The configuration consists of a rectangular channel with a constant-height cross section, where actuation is applied through spanwise-oriented blowing and suction jets (indicated by the orange plane) along the bottom wall. Sensor measurements are sampled at a distance of $y^+ = 15$ from the wall at locations directly above the actuator (shown by the pink plane). A smaller channel variant shares the same height but has half the streamwise length and spanwise depth. In the *bottom-actuation* variant, only the sensor observations from the bottom wall are provided to the control policy. The actuator visualizations are not drawn to scale and do not represent the actual number of control units; they are shown purely for illustration. In the small channel, $32 \times 32$ actuators are placed per wall, whereas in the large channel $64 \times 64$ actuators are used.

**Reward Function**    The reward is defined based on the reduction of instantaneous wall shear stress $r_t = 1 - \tau_{\mathrm{wall}}/\tau_{\mathrm{wall,ref}}$, where $\tau_{\mathrm{wall,ref}}$ is the reference value of the uncontrolled flow. The wall shear stress is computed as

$$\tau_{wall} = \nu \left. \frac{\partial u_x}{\partial y} \right|_{y=0}. \tag{7}$$

For environments with single-wall actuation, only the bottom wall is considered; for dual-wall actuation, the stress is averaged across both walls.

**Actuation**    The control is applied via wall-normal blowing and suction at the boundary using multiple spatially distributed actuators, where a zero net-mass-flux is enforced. The boundary velocities are scaled to enforce a maximum value equal to the friction velocity $u_+$. Two configurations are provided: one with actuation at the bottom wall only, and one with actuation at both walls.

**Observations**    Observations consist of the local velocity fluctuations $\mathbf{u}' = (u', v')$, defined as deviations from the instantaneous spatial mean velocity. Specifically, the velocity components are decomposed as

$$u' = u - \langle u \rangle_V, \qquad v' = v - \langle v \rangle_V, \tag{8}$$

where $\langle \cdot \rangle_V$ denotes volumetric averaging. The fluctuations are sampled at a detection plane located at a wall-normal distance $y_+ = 15$, directly above the corresponding wall actuator.

**Difficulty Levels**  Difficulty is defined using the friction Reynolds number ($\mathrm{Re}_\tau$). We use `easy` at $\mathrm{Re}_\tau = 180$, `medium` at $\mathrm{Re}_\tau = 330$, and `hard` at $\mathrm{Re}_\tau = 550$.

**Opposition Control Baseline**  For the TCF, a commonly used baseline control strategy is opposition control (Guastoni et al., 2023; Sonoda et al., 2023). In this approach, wall-normal blowing and suction are applied in opposition to the sensed wall-normal velocity fluctuations. Specifically, the wall-normal velocity at the wall is prescribed as

$$v_{\mathrm{wall}}(x, z, t) = -\alpha\, v'(x, y_s, z, t), \tag{9}$$

where $v'(x, y_s, z, t)$ denotes the wall-normal velocity fluctuation measured at the detection plane located at $y_s$ (corresponding to $y_+ = 15$), and $\alpha > 0$ is a control gain, set to $1.0$ in this work.

## D. Experimental Setup

### D.1. Hardware and Software Configuration

**General Experimental Setup**  Unless stated otherwise, all experiments were conducted using the following shared hardware and software configuration:

- **Python:** 3.10

- **PyTorch:** 2.9.1

- **CUDA:** 12.8

- **System Memory:** 32 GB RAM

- **CPU:** 32 cores of an AMD EPYC 7742 (64-core processor)

- **GPU:** $1 \times$ NVIDIA A100 (40 GB or 80 GB)

**CylinderJet3D-hard-v0 Environment**  Experiments for the `CylinderJet3D-hard-v0` environment and SARL were conducted on compute nodes with the following differing hardware configurations:

- **CPU:** 8 cores of an AMD EPYC 7763 (Milan architecture)

- **GPU:** $2 \times$ NVIDIA A100 (40 GB)

### D.2. Algorithm Hyperparameter Configurations

The hyperparameters used in our experiments are stated in Tables 5 and 6 for PPO and SAC, respectively. We note that for all SAC experiments on TCF environments, we set the number of gradient steps per update to 1 to avoid excessive gradient updates due to the large number of pseudo multi-agent environments.

For MA-PPO and MA-SAC, each individual actor corresponds to one pseudo-environment.

### D.3. Differentiable Predictive Control

To isolate the value of reward gradients in our fully differentiable AFC benchmark, we evaluate a differentiable predictive control (DPC; Drgoňa et al. (2022)) baseline that relies solely on gradient information provided by FluidGym. Notably, instead of predicting a sequence of actions given a history of observations, we adapt DPC to the RL setup, i.e., predicting $a_t$ given $o_t$. Starting at $o_0$, DPC performs a rollout over a horizon of $H$ steps and collects the rewards without detaching them from the computational graph. After the last step, the rewards are accumulated as the discounted return. This leads to the loss function $\mathcal{L}_{\mathrm{DPC}} = -\sum_{t=0}^{H-1} \gamma^t r_t$, which is backpropagated through time to the policy parameters and updated using the Adam optimizer (Kingma & Ba, 2015) with learning rate $\alpha$. To stabilize training, we constrain the gradient magnitude by setting a global clipping threshold $\omega$. In our experiments, we set $H = 40$, $\gamma = 0.999$, $\alpha = 1 \cdot 10^3$, and $\omega = 0.5$. We note that for `CylinderJet2D-easy-v0`, the DPC training degraded after 10k steps and was therefore stopped after 10k instead of 50k steps.

*Table 5.* PPO hyperparameters used in all experiments.

| HYPERPARAMETER | VALUE |
|---|---|
| POLICY NETWORK TYPE | MLPPOLICY |
| POLICY NETWORK LAYERS | [64, 64] |
| CRITIC NETWORK LAYERS | [64, 64] |
| LEARNING RATE | $3 \times 10^{-4}$ |
| STEPS PER ROLLOUT ($n\_steps$) | 2048 |
| BATCH SIZE | 64 |
| UPDATE EPOCHS ($n\_epochs$) | 10 |
| DISCOUNT FACTOR ($\gamma$) | 0.99 |
| GAE $\lambda$ | 0.95 |
| CLIP RANGE | 0.2 |
| ADVANTAGE NORMALIZATION | TRUE |
| ENTROPY COEFFICIENT ($c\_$ent) | 0.01 |
| VALUE FUNCTION COEFFICIENT ($c\_$vf) | 0.5 |
| MAX GRADIENT NORM | 0.5 |
| DEVICE | CPU |

*Table 6.* SAC hyperparameters used in all experiments except TCF environments. For TCF, we set the number of gradient steps per update to 1.

| HYPERPARAMETER | VALUE |
|---|---|
| POLICY NETWORK TYPE | MLPPOLICY |
| POLICY NETWORK LAYERS | [256, 256] |
| CRITIC NETWORK LAYERS | [256, 256] |
| LEARNING RATE | $3 \times 10^{-4}$ |
| DISCOUNT FACTOR ($\gamma$) | 0.99 |
| SOFT UPDATE COEFFICIENT ($\tau$) | 0.005 |
| REPLAY BUFFER SIZE | $10^{6}$ |
| BATCH SIZE | 256 |
| LEARNING STARTS | 100 |
| TRAINING FREQUENCY | 1 |
| GRADIENT STEPS PER UPDATE | $-1$ (EQUAL TO TRAIN_FREQ) |
| ENTROPY COEFFICIENT ($\alpha$) | AUTO |
| TARGET ENTROPY | AUTO |
| TARGET UPDATE INTERVAL | 1 |
| DEVICE | CUDA |

*Table 7.* TD-MPC hyperparameters used in all experiments.

| HYPERPARAMETER | VALUE |
|---|---|
| MODALITY | STATE |
| ACTION REPEAT | 1 |
| TOTAL TRAINING STEPS ($T$) | $T$ |
| SEED STEPS | 100 |
| UPDATE FREQUENCY | 2 |
| ITERATIONS | 6 |
| NUMBER OF SAMPLES | 512 |
| NUMBER OF ELITES | 64 |
| MIXTURE COEFFICIENT | 0.05 |
| MINIMUM STD. DEV. | 0.05 |
| TEMPERATURE | 0.5 |
| MOMENTUM | 0.1 |
| DISCOUNT FACTOR ($\gamma$) | 0.99 |
| BATCH SIZE | 512 |
| MAX BUFFER SIZE | $T$ |
| REWARD COEFFICIENT | 0.5 |
| VALUE COEFFICIENT | 0.1 |
| CONSISTENCY COEFFICIENT | 2 |
| $\rho$ | 0.5 |
| $\kappa$ | 0.1 |
| LEARNING RATE | $1 \times 10^{-3}$ |
| TARGET NETWORK UPDATE ($\tau$) | 0.01 |
| MAX GRADIENT NORM | 10 |
| STD. DEV. SCHEDULE | LINEAR$(0.5, 0.05, T)$ |
| HORIZON SCHEDULE | LINEAR$(1, 20, T)$ |
| PRIORITIZED EXP. REPLAY ($\alpha, \beta$) | 0.6, 0.4 |
| ENCODER DIMENSION | 256 |
| MLP DIMENSION | 512 |
| LATENT DIMENSION | 50 |

# E. Additional Results

## E.1. Runtime Benchmarks

*Table 8.* Experiment details and GPU runtimes. Total GPU hours are computed as #steps × #seeds × #seconds × #algorithms. Entries with zero GPU hours correspond to environment variants (e.g., wider domains or alternative actuation layouts) that were not evaluated separately, since results from the corresponding base configurations are representative.

| ENVIRONMENT | DIFFICULTY | #STEPS | #SEEDS | SECONDS PER STEP | #ALGORITHMS | GPU HOURS |
|---|---|---|---|---|---|---|
| CYLINDERROT2D | EASY | 50000 | 5 | 1.241 | 2 | 172.40 |
| CYLINDERROT2D | MEDIUM | 50000 | 5 | 2.059 | 2 | 285.96 |
| CYLINDERROT2D | HARD | 50000 | 5 | 2.561 | 2 | 355.70 |
| CYLINDERJET2D | EASY | 50000 | 5 | 1.259 | 2 | 174.89 |
| CYLINDERJET2D | MEDIUM | 50000 | 5 | 2.209 | 2 | 306.74 |
| CYLINDERJET2D | HARD | 50000 | 5 | 2.552 | 2 | 354.43 |
| CYLINDERJET3D | EASY | 50000 | 5 | 4.209 | 4 | 1169.30 |
| CYLINDERJET3D | MEDIUM | 50000 | 5 | 7.684 | 4 | 2134.52 |
| CYLINDERJET3D | HARD | 50000 | 5 | 16.679 | 4 | 4633.18 |
| RBC2D | EASY | 50000 | 5 | 1.265 | 4 | 351.26 |
| RBC2D | MEDIUM | 50000 | 5 | 2.232 | 4 | 620 |
| RBC2D | HARD | 50000 | 5 | 2.260 | 4 | 627.73 |
| RBC2D-WIDE | EASY | 50000 | 5 | 1.314 | 4 | 0.00 |
| RBC2D-WIDE | MEDIUM | 50000 | 5 | 2.292 | 4 | 0.00 |
| RBC2D-WIDE | HARD | 50000 | 5 | 2.349 | 4 | 0.00 |
| RBC3D | EASY | 50000 | 5 | 1.168 | 2 | 162.25 |
| RBC3D | MEDIUM | 50000 | 5 | 1.157 | 2 | 160.73 |
| RBC3D | HARD | 50000 | 5 | 1.199 | 2 | 166.57 |
| RBC3D-WIDE | EASY | 50000 | 5 | 1.675 | 2 | 0.00 |
| RBC3D-WIDE | MEDIUM | 50000 | 5 | 1.689 | 2 | 0.00 |
| RBC3D-WIDE | HARD | 50000 | 5 | 1.754 | 2 | 0.00 |
| AIRFOIL2D | EASY | 20000 | 5 | 18.851 | 2 | 1047.29 |
| AIRFOIL2D | MEDIUM | 20000 | 5 | 30.145 | 2 | 1674.74 |
| AIRFOIL2D | HARD | 20000 | 5 | 37.278 | 2 | 2071.01 |
| AIRFOIL3D | EASY | 20000 | 5 | 34.526 | 4 | 3836.27 |
| AIRFOIL3D | MEDIUM | 20000 | 5 | 59.840 | 3 | 4986.65 |
| AIRFOIL3D | HARD | 10000 | 5 | 63.153 | 1 | 877.120 |
| TCFSMALL3D-BOTH | EASY | 100000 | 5 | 0.481 | 2 | 133.68 |
| TCFSMALL3D-BOTH | MEDIUM | 100000 | 5 | 0.250 | 2 | 69.38 |
| TCFSMALL3D-BOTH | HARD | 100000 | 5 | 0.248 | 2 | 68.94 |
| TCFSMALL3D-BOTTOM | EASY | 100000 | 5 | 0.427 | 2 | 0.00 |
| TCFSMALL3D-BOTTOM | MEDIUM | 100000 | 5 | 0.218 | 2 | 0.00 |
| TCFSMALL3D-BOTTOM | HARD | 100000 | 5 | 0.220 | 2 | 0.00 |
| TCFLARGE3D-BOTH | EASY | 100000 | 5 | 0.846 | 2 | 235.06 |
| TCFLARGE3D-BOTH | MEDIUM | 100000 | 5 | 0.417 | 2 | 115.92 |
| TCFLARGE3D-BOTH | HARD | 100000 | 5 | 0.417 | 2 | 115.78 |
| TCFLARGE3D-BOTTOM | EASY | 100000 | 5 | 0.759 | 2 | 0.00 |
| TCFLARGE3D-BOTTOM | MEDIUM | 100000 | 5 | 0.387 | 2 | 0.00 |
| TCFLARGE3D-BOTTOM | HARD | 100000 | 5 | 0.408 | 2 | 0.00 |
| **TOTAL** | | | | | | **26 907.50** |

Table 8 states individual environment wall-clock times per step as well as the number of steps, seeds, and total GPU hours of the experiments presented in this paper. Results were obtained by running 80 (8 for medium and hard 3D airfoil cases) RL steps with random actions and averaging the results. Experiments were conducted on a single NVIDIA A100 GPU.

## E.2. Quantitative Training Results

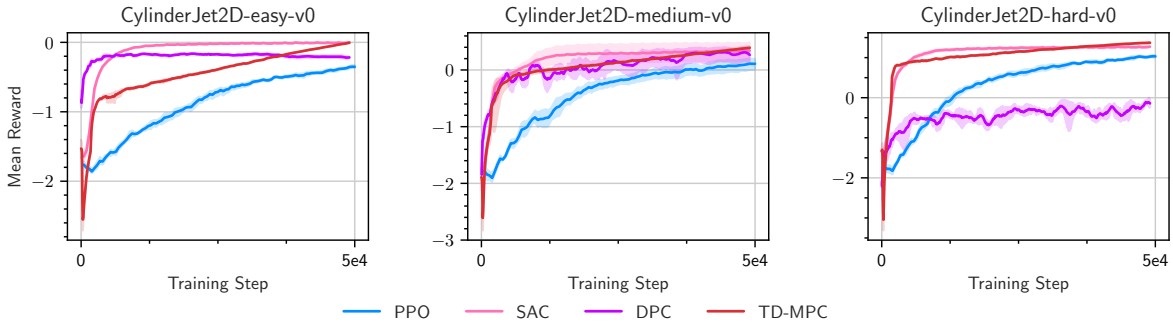

*Figure 18.* Mean training reward for `CylinderJet2D`. Error bars indicate 95% confidence intervals.

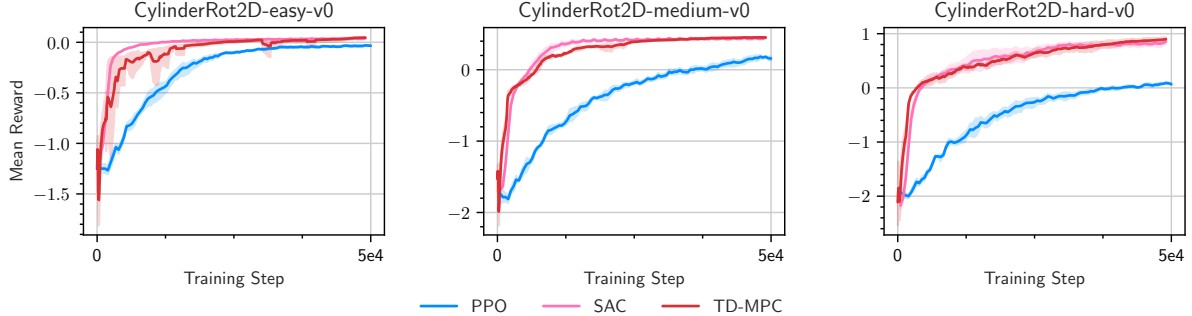

*Figure 19.* Mean training reward for `CylinderRot2D`. Error bars indicate 95% confidence intervals.

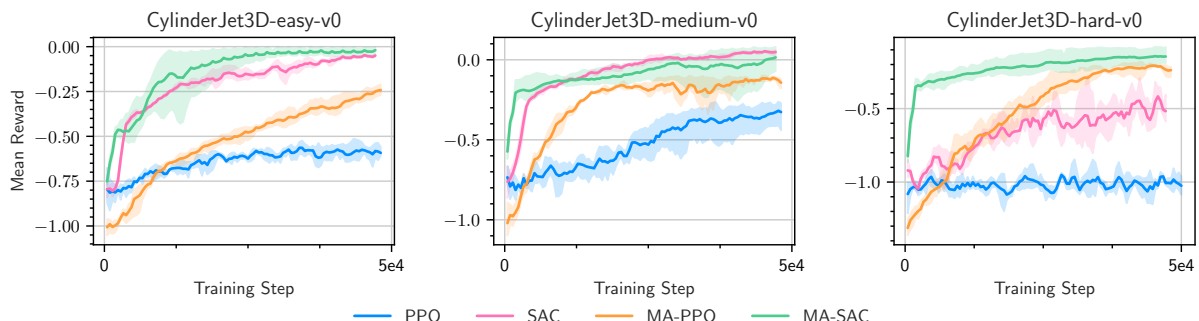

*Figure 20.* Mean training reward for `CylinderJet3D`. Error bars indicate 95% confidence intervals.

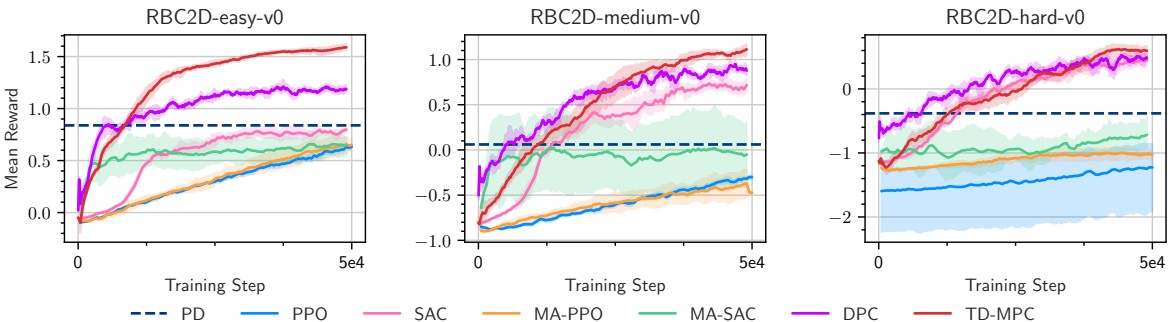

*Figure 21.* Mean training reward for `RBC2D`. Error bars indicate 95% confidence intervals.

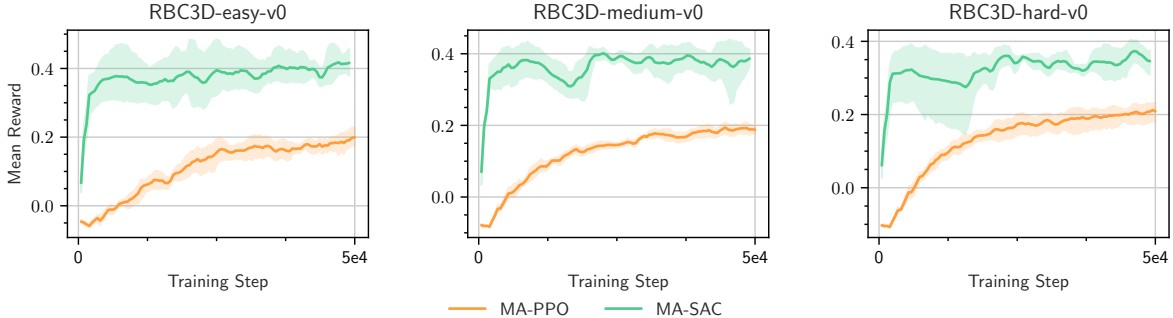

*Figure 22.* Mean training reward for `RBC3D`. Error bars indicate 95% confidence intervals.

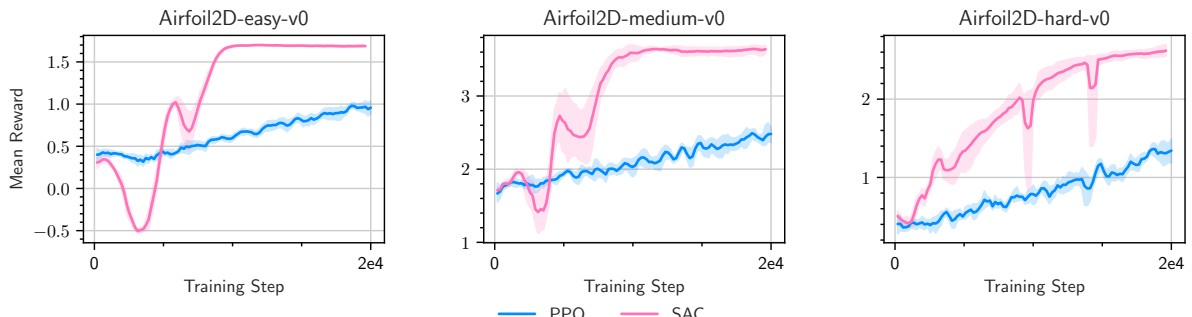

*Figure 23.* Mean training reward for `Airfoil2D`. Error bars indicate 95% confidence intervals.

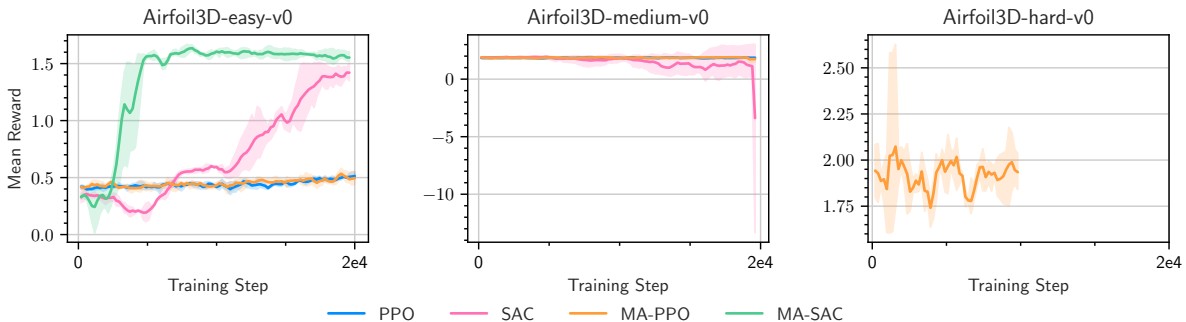

*Figure 24.* Mean training reward for `Airfoil3D`. Error bars indicate 95% confidence intervals.

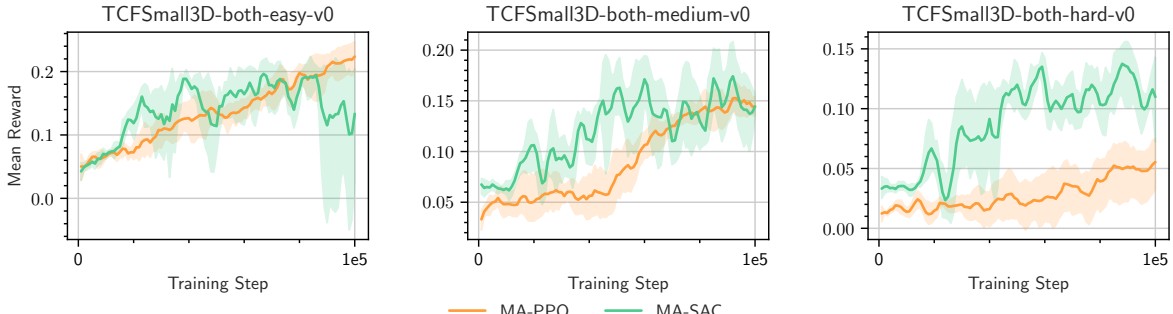

*Figure 25.* Mean training reward for `TCFSmall3D-both`. Error bars indicate 95% confidence intervals.

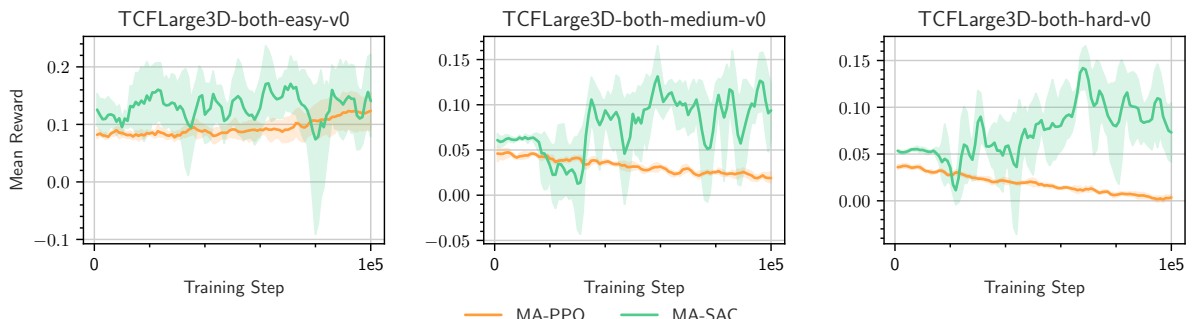

*Figure 26.* Mean training reward for `TCFLarge3D-both`. Error bars indicate 95% confidence intervals.

*Table 9.* Cylinder test set metrics. All values report the interquartile mean (IQM) over test episodes and random seeds. Drag reduction is measured relative to the mean drag over 10 uncontrolled training episodes of the baseflow. Best result per environment is highlighted in **bold**.

| ENVIRONMENT | ALGORITHM | REWARD | $C_D$ | $C_L$ | DRAG REDUCTION (%) |
|---|---|---|---|---|---|
| CYLINDERROT2D-EASY-V0 | BASEFLOW | - | 3.328 | −0.042 | - |
| CYLINDERROT2D-EASY-V0 | PPO | −0.002 | 3.191 | 0.035 | 4.125 |
| CYLINDERROT2D-EASY-V0 | SAC | 0.037 | 3.179 | 0.016 | 4.477 |
| CYLINDERROT2D-EASY-V0 | TD-MPC | 0.050 | 3.170 | 0.006 | **4.747** |
| CYLINDERROT2D-MEDIUM-V0 | BASEFLOW | - | 3.152 | 0.037 | - |
| CYLINDERROT2D-MEDIUM-V0 | PPO | 0.309 | 2.489 | −0.060 | 21.033 |
| CYLINDERROT2D-MEDIUM-V0 | SAC | 0.344 | 2.475 | −0.093 | 21.496 |
| CYLINDERROT2D-MEDIUM-V0 | TD-MPC | 0.424 | 2.447 | −0.016 | **22.362** |
| CYLINDERROT2D-HARD-V0 | BASEFLOW | - | 3.619 | 0.057 | - |
| CYLINDERROT2D-HARD-V0 | PPO | 0.162 | 2.962 | −0.028 | 18.162 |
| CYLINDERROT2D-HARD-V0 | SAC | 0.552 | 2.440 | −0.180 | 32.587 |
| CYLINDERROT2D-HARD-V0 | TD-MPC | 0.493 | 2.387 | −0.081 | **34.056** |
| CYLINDERJET2D-EASY-V0 | BASEFLOW | - | 3.328 | −0.042 | - |
| CYLINDERJET2D-EASY-V0 | PPO | −0.052 | 3.141 | 0.065 | 5.638 |
| CYLINDERJET2D-EASY-V0 | SAC | 0.051 | 3.105 | 0.032 | **6.697** |
| CYLINDERJET2D-EASY-V0 | DPC | −0.149 | 3.184 | 0.090 | 4.335 |
| CYLINDERJET2D-EASY-V0 | TD-MPC | 0.024 | 3.116 | 0.036 | 6.360 |
| CYLINDERJET2D-MEDIUM-V0 | BASEFLOW | - | 3.152 | 0.037 | - |
| CYLINDERJET2D-MEDIUM-V0 | PPO | 0.274 | 2.487 | −0.066 | 21.110 |
| CYLINDERJET2D-MEDIUM-V0 | SAC | 0.426 | 2.484 | −0.004 | **21.216** |
| CYLINDERJET2D-MEDIUM-V0 | DPC | −0.150 | 2.836 | −0.299 | 10.039 |
| CYLINDERJET2D-MEDIUM-V0 | TD-MPC | 0.405 | 2.505 | −0.027 | 20.547 |
| CYLINDERJET2D-HARD-V0 | BASEFLOW | - | 3.619 | 0.057 | - |
| CYLINDERJET2D-HARD-V0 | PPO | 1.173 | 2.158 | 0.038 | 40.385 |
| CYLINDERJET2D-HARD-V0 | SAC | 1.352 | 2.011 | 0.001 | **44.426** |
| CYLINDERJET2D-HARD-V0 | DPC | −0.336 | 2.652 | 0.524 | 26.722 |
| CYLINDERJET2D-HARD-V0 | TD-MPC | 1.350 | 2.015 | 0.062 | 44.320 |
| CYLINDERJET3D-EASY-V0 | BASEFLOW | - | 3.305 | −0.028 | - |
| CYLINDERJET3D-EASY-V0 | PPO | −0.232 | 3.194 | −0.021 | 3.376 |
| CYLINDERJET3D-EASY-V0 | SAC | −0.040 | 3.222 | 0.032 | 2.531 |
| CYLINDERJET3D-EASY-V0 | MA-PPO | −0.203 | 3.198 | 0.031 | 3.254 |
| CYLINDERJET3D-EASY-V0 | MA-SAC | −0.058 | 3.109 | 0.001 | **5.933** |
| CYLINDERJET3D-MEDIUM-V0 | BASEFLOW | - | 2.984 | −0.008 | - |
| CYLINDERJET3D-MEDIUM-V0 | PPO | −0.211 | 2.770 | −0.194 | 7.194 |
| CYLINDERJET3D-MEDIUM-V0 | SAC | 0.010 | 2.797 | 0.025 | 6.273 |
| CYLINDERJET3D-MEDIUM-V0 | MA-PPO | −0.239 | 2.930 | 0.044 | 1.810 |
| CYLINDERJET3D-MEDIUM-V0 | MA-SAC | 0.031 | 2.729 | 0.022 | **8.556** |
| CYLINDERJET3D-HARD-V0 | BASEFLOW | - | 2.571 | −0.018 | - |
| CYLINDERJET3D-HARD-V0 | PPO | −0.727 | 2.547 | −0.044 | 0.945 |
| CYLINDERJET3D-HARD-V0 | SAC | −1.441 | 2.644 | 0.144 | −2.814 |
| CYLINDERJET3D-HARD-V0 | MA-PPO | −1.043 | 2.569 | 0.030 | 0.091 |
| CYLINDERJET3D-HARD-V0 | MA-SAC | −0.941 | 2.511 | 0.018 | **2.352** |

*Table 10.* RBC test set metrics. All values report the interquartile mean (IQM) over test episodes and random seeds. Heat transfer improvement is measured relative to the mean instant Nusselt number $\mathrm{Nu_{instant}}$ over 10 uncontrolled training episodes of the baseflow. Best result per environment is highlighted in **bold**.

| ENVIRONMENT | ALGORITHM | REWARD | $\mathrm{Nu_{instant}}$ | HEAT TRANSFER IMPROVEMENT (%) |
|---|---|---|---|---|
| RBC2D-EASY-V0 | BASEFLOW | - | 4.841 | - |
| RBC2D-EASY-V0 | PD | 0.838 | 4.057 | 16.187 |
| RBC2D-EASY-V0 | PPO | 0.888 | 4.008 | 17.200 |
| RBC2D-EASY-V0 | SAC | 0.779 | 4.117 | 14.952 |
| RBC2D-EASY-V0 | MA-PPO | 1.024 | 3.872 | 20.015 |
| RBC2D-EASY-V0 | MA-SAC | 0.650 | 4.246 | 12.285 |
| RBC2D-EASY-V0 | DPC | 1.254 | 3.642 | 24.770 |
| RBC2D-EASY-V0 | TD-MPC | 1.587 | 3.309 | **31.650** |
| RBC2D-MEDIUM-V0 | BASEFLOW | - | 6.856 | - |
| RBC2D-MEDIUM-V0 | PD | 0.061 | 6.368 | 7.112 |
| RBC2D-MEDIUM-V0 | PPO | 0.138 | 6.291 | 8.238 |
| RBC2D-MEDIUM-V0 | SAC | 0.790 | 5.639 | 17.746 |
| RBC2D-MEDIUM-V0 | MA-PPO | 0.056 | 6.373 | 7.041 |
| RBC2D-MEDIUM-V0 | MA-SAC | −0.018 | 6.447 | 5.960 |
| RBC2D-MEDIUM-V0 | DPC | 0.993 | 5.436 | 20.706 |
| RBC2D-MEDIUM-V0 | TD-MPC | 1.070 | 5.359 | **21.833** |
| RBC2D-HARD-V0 | BASEFLOW | - | 7.854 | - |
| RBC2D-HARD-V0 | PD | −0.382 | 7.624 | 2.919 |
| RBC2D-HARD-V0 | PPO | −0.304 | 7.547 | 3.911 |
| RBC2D-HARD-V0 | SAC | 0.525 | 6.717 | 14.467 |
| RBC2D-HARD-V0 | MA-PPO | −0.484 | 7.726 | 1.622 |
| RBC2D-HARD-V0 | MA-SAC | −0.715 | 7.958 | −1.327 |
| RBC2D-HARD-V0 | DPC | 0.637 | 6.606 | 15.890 |
| RBC2D-HARD-V0 | TD-MPC | 0.653 | 6.589 | **16.099** |
| RBC3D-EASY-V0 | BASEFLOW | - | 2.182 | - |
| RBC3D-EASY-V0 | MA-PPO | 0.367 | 1.815 | 16.815 |
| RBC3D-EASY-V0 | MA-SAC | 0.400 | 1.782 | **18.333** |
| RBC3D-MEDIUM-V0 | BASEFLOW | - | 2.444 | - |
| RBC3D-MEDIUM-V0 | MA-PPO | 0.340 | 2.105 | 13.893 |
| RBC3D-MEDIUM-V0 | MA-SAC | 0.384 | 2.061 | **15.692** |
| RBC3D-HARD-V0 | BASEFLOW | - | 2.684 | - |
| RBC3D-HARD-V0 | MA-PPO | 0.341 | 2.343 | **12.713** |
| RBC3D-HARD-V0 | MA-SAC | 0.323 | 2.361 | 12.050 |

*Table 11.* Airfoil test set metrics. All values report the interquartile mean (IQM) over test episodes and random seeds. Aerodynamic efficiency improvement is measured relative to the mean aerodynamic efficiency over 10 uncontrolled training episodes of the baseflow. Best result per environment is highlighted in **bold**.

| ENVIRONMENT | ALGORITHM | REWARD | AERODYNAMIC EFFICIENCY | IMPROVEMENT (%) |
|---|---|---|---|---|
| AIRFOIL2D-EASY-V0 | BASEFLOW | - | 2.887 | - |
| AIRFOIL2D-EASY-V0 | PPO | 1.422 | 4.309 | 49.265 |
| AIRFOIL2D-EASY-V0 | SAC | 1.705 | 4.592 | **59.072** |
| AIRFOIL2D-MEDIUM-V0 | BASEFLOW | - | 3.572 | - |
| AIRFOIL2D-MEDIUM-V0 | PPO | 3.134 | 6.706 | 87.747 |
| AIRFOIL2D-MEDIUM-V0 | SAC | 3.666 | 7.238 | **102.633** |
| AIRFOIL2D-HARD-V0 | BASEFLOW | - | 6.063 | - |
| AIRFOIL2D-HARD-V0 | PPO | 1.338 | 7.401 | 22.065 |
| AIRFOIL2D-HARD-V0 | SAC | 2.636 | 8.699 | **43.470** |
| AIRFOIL3D-EASY-V0 | BASEFLOW | - | 2.838 | - |
| AIRFOIL3D-EASY-V0 | PPO | $-0.105$ | 2.733 | $-3.691$ |
| AIRFOIL3D-EASY-V0 | SAC | 1.462 | 4.300 | 51.513 |
| AIRFOIL3D-EASY-V0 | MA-PPO | 0.084 | 2.922 | 2.951 |
| AIRFOIL3D-EASY-V0 | MA-SAC | 1.584 | 4.422 | **55.808** |
| AIRFOIL3D-MEDIUM-V0 | BASEFLOW | - | 3.589 | - |
| AIRFOIL3D-MEDIUM-V0 | PPO | 0.126 | 3.715 | 3.500 |
| AIRFOIL3D-MEDIUM-V0 | SAC | 0.823 | 4.412 | **22.928** |
| AIRFOIL3D-MEDIUM-V0 | MA-PPO | 0.025 | 3.614 | 0.707 |
| AIRFOIL3D-HARD-V0 | BASEFLOW | - | 4.965 | - |
| AIRFOIL3D-HARD-V0 | MA-PPO | 0.433 | 5.398 | **8.715** |

*Table 12.* TCF test set metrics. All values report the interquartile mean (IQM) over test episodes and random seeds. Drag reduction is measured relative to the mean wall stress $\tau_{\text{wall}}$ over 10 uncontrolled training episodes of the baseflow. Best result per environment is highlighted in **bold**.

| ENVIRONMENT | ALGORITHM | REWARD | $\tau_{\text{wall}}$ | DRAG REDUCTION (%) |
|---|---|---|---|---|
| TCFSMALL3D-BOTH-EASY-V0 | BASEFLOW | - | 0.002 | - |
| TCFSMALL3D-BOTH-EASY-V0 | MA-PPO | 0.207 | 0.001 | **20.689** |
| TCFSMALL3D-BOTH-EASY-V0 | MA-SAC | 0.171 | 0.001 | 17.091 |
| TCFSMALL3D-BOTH-MEDIUM-V0 | BASEFLOW | - | 0.002 | - |
| TCFSMALL3D-BOTH-MEDIUM-V0 | MA-PPO | 0.193 | 0.001 | **19.281** |
| TCFSMALL3D-BOTH-MEDIUM-V0 | MA-SAC | 0.173 | 0.001 | 17.290 |
| TCFSMALL3D-BOTH-HARD-V0 | BASEFLOW | - | 0.001 | - |
| TCFSMALL3D-BOTH-HARD-V0 | MA-PPO | 0.120 | 0.001 | **11.999** |
| TCFSMALL3D-BOTH-HARD-V0 | MA-SAC | 0.089 | 0.001 | 8.945 |
| TCFLARGE3D-BOTH-EASY-V0 | BASEFLOW | - | 0.002 | - |
| TCFLARGE3D-BOTH-EASY-V0 | MA-PPO | 0.129 | 0.002 | **12.885** |
| TCFLARGE3D-BOTH-EASY-V0 | MA-SAC | 0.045 | 0.002 | 4.514 |
| TCFLARGE3D-BOTH-MEDIUM-V0 | BASEFLOW | - | 0.002 | - |
| TCFLARGE3D-BOTH-MEDIUM-V0 | MA-PPO | 0.019 | 0.002 | 1.903 |
| TCFLARGE3D-BOTH-MEDIUM-V0 | MA-SAC | 0.094 | 0.001 | **9.415** |
| TCFLARGE3D-BOTH-HARD-V0 | BASEFLOW | - | 0.001 | - |
| TCFLARGE3D-BOTH-HARD-V0 | MA-PPO | 0.001 | 0.001 | 0.113 |
| TCFLARGE3D-BOTH-HARD-V0 | MA-SAC | 0.059 | 0.001 | **5.877** |

### E.3. Quantitative Test Results

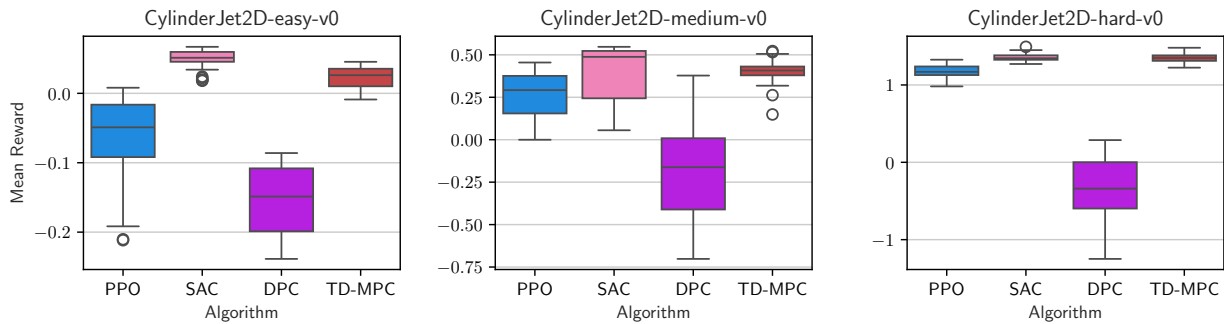

*Figure 27.* Mean test reward for `CylinderJet2D`.

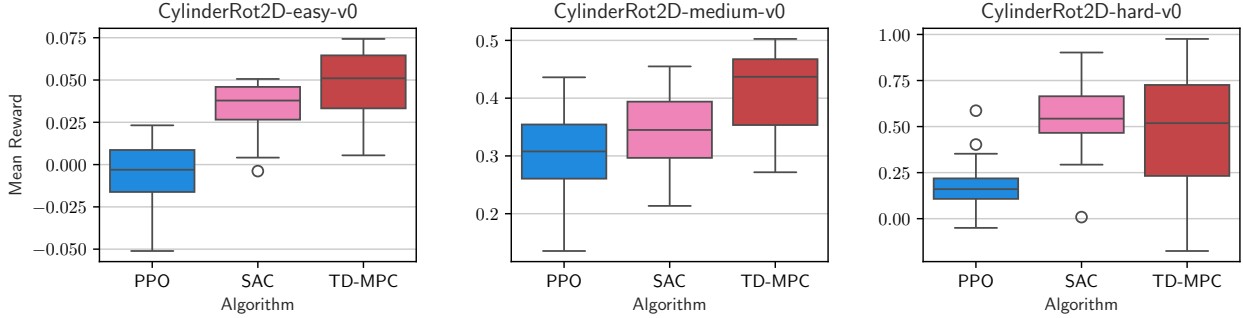

*Figure 28.* Mean test reward for `CylinderRot2D`.

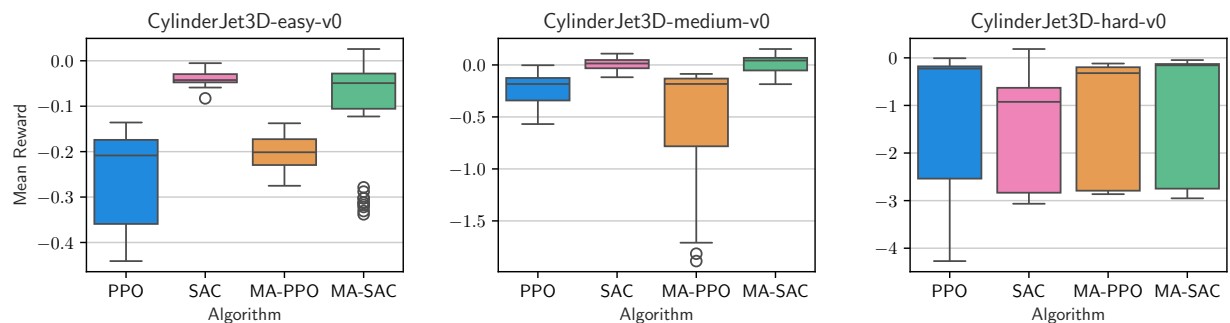

*Figure 29.* Mean test reward for `CylinderJet3D`.

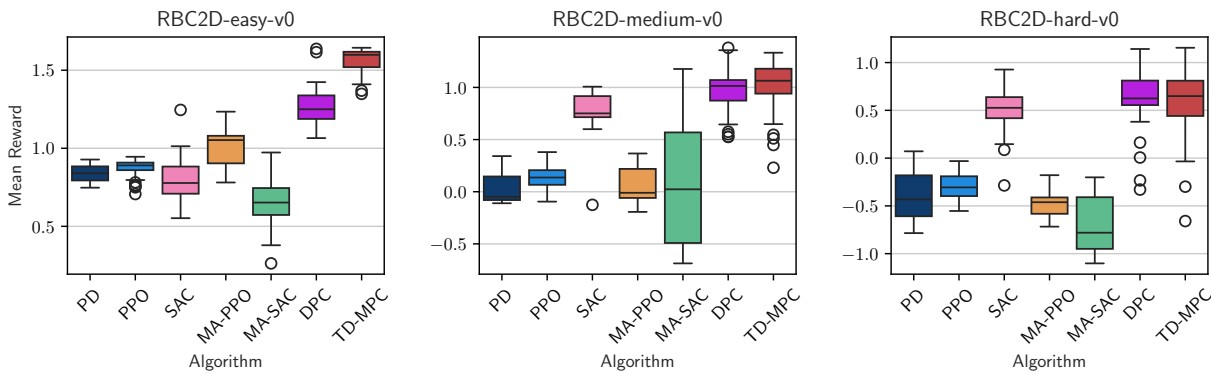

*Figure 30.* Mean test reward for `RBC2D`.

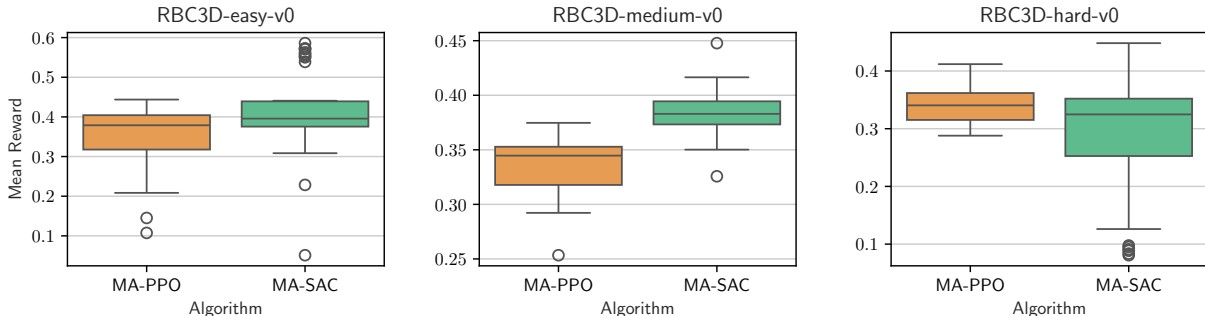

*Figure 31.* Mean test reward for `RBC3D`.

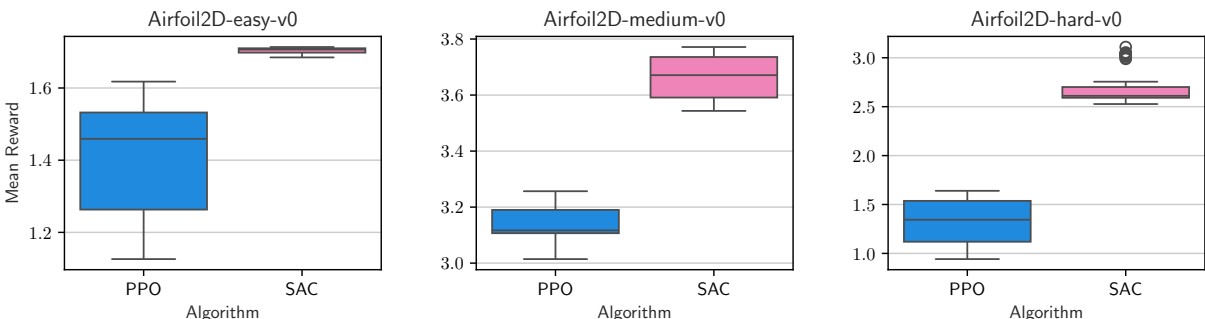

*Figure 32.* Mean test reward for `Airfoil2D`.

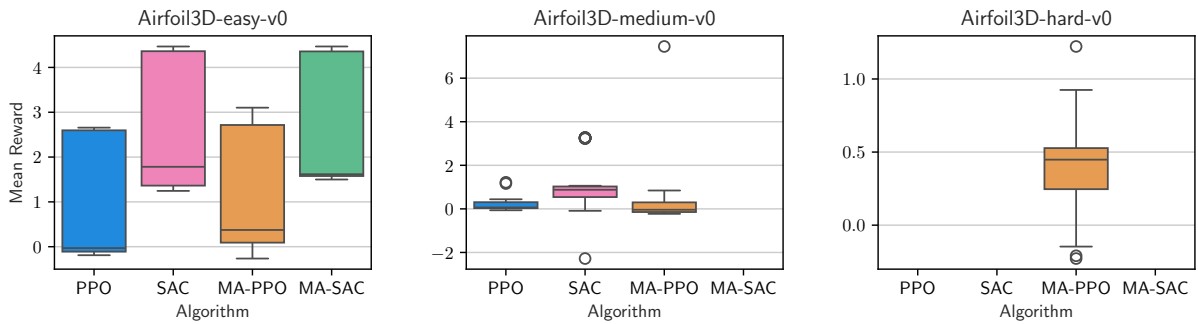

*Figure 33.* Mean test reward for `Airfoil3D`. We note that due to computational constraints, MA-PPO results on the hard case are currently limited to four instead of five seeds.

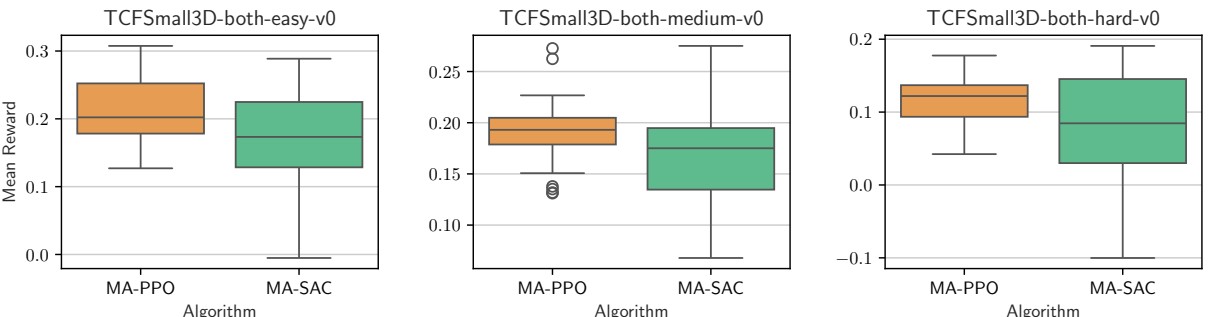

*Figure 34.* Mean test reward for `TCFSmall3D-both`.

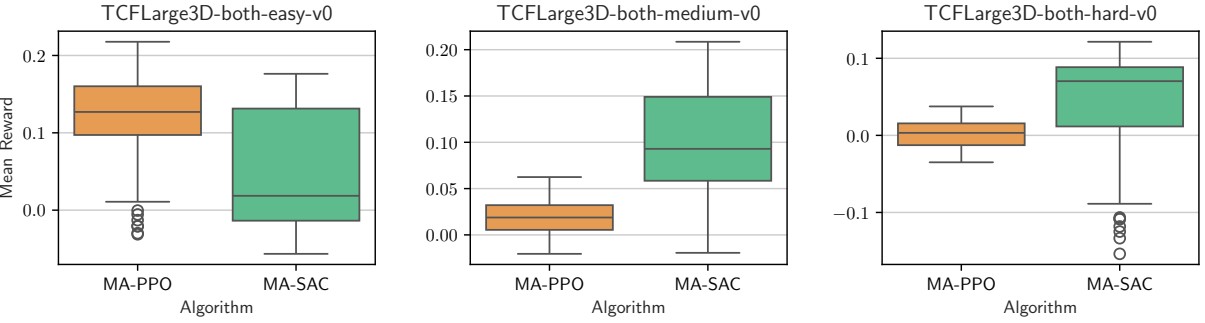

*Figure 35.* Mean test reward for `TCFLarge3D-both`.

## E.4. Qualitative Test Results

In the following, we present qualitative visualizations of uncontrolled and final controlled flow fields for all environments and algorithms for seed 0.

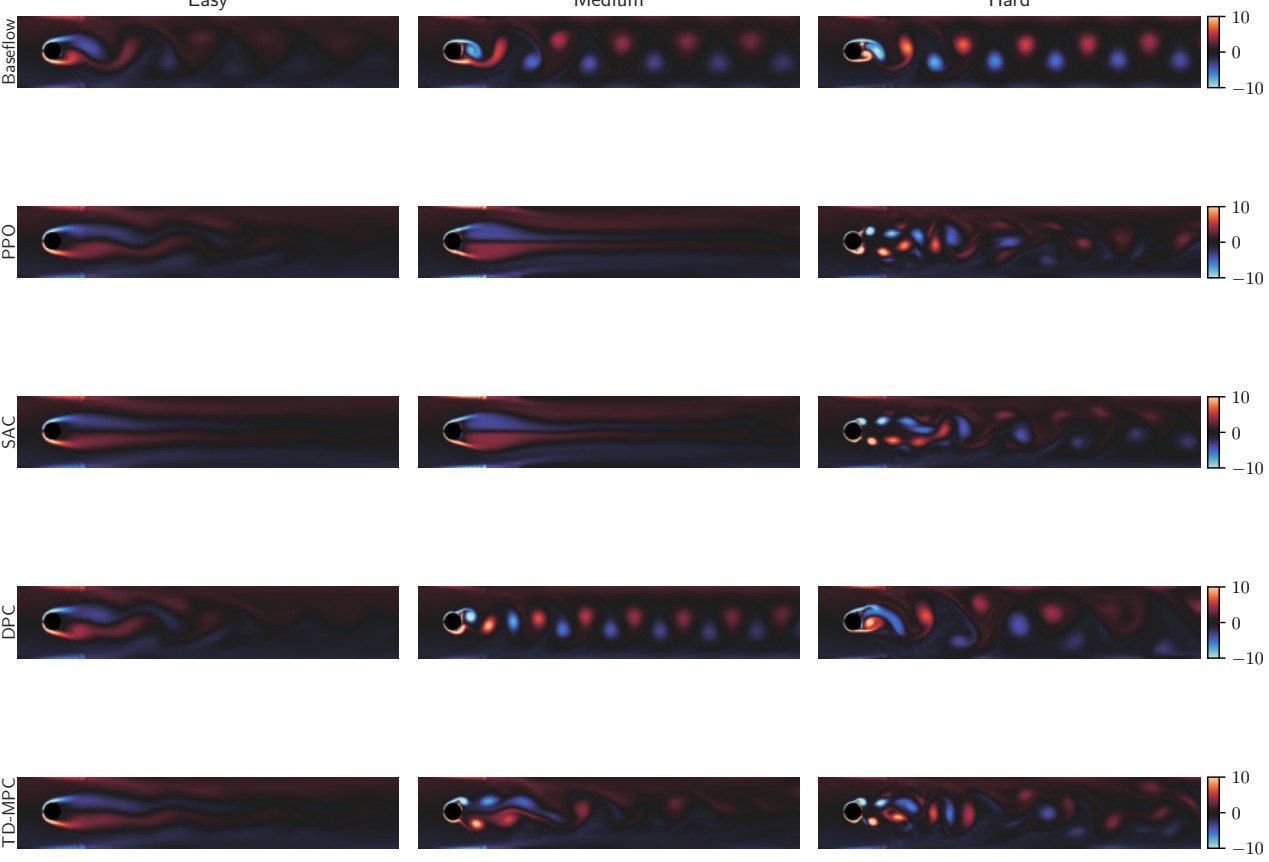

*Figure 36.* Qualitative test results for `CylinderJet2D`, showing the vorticity field.

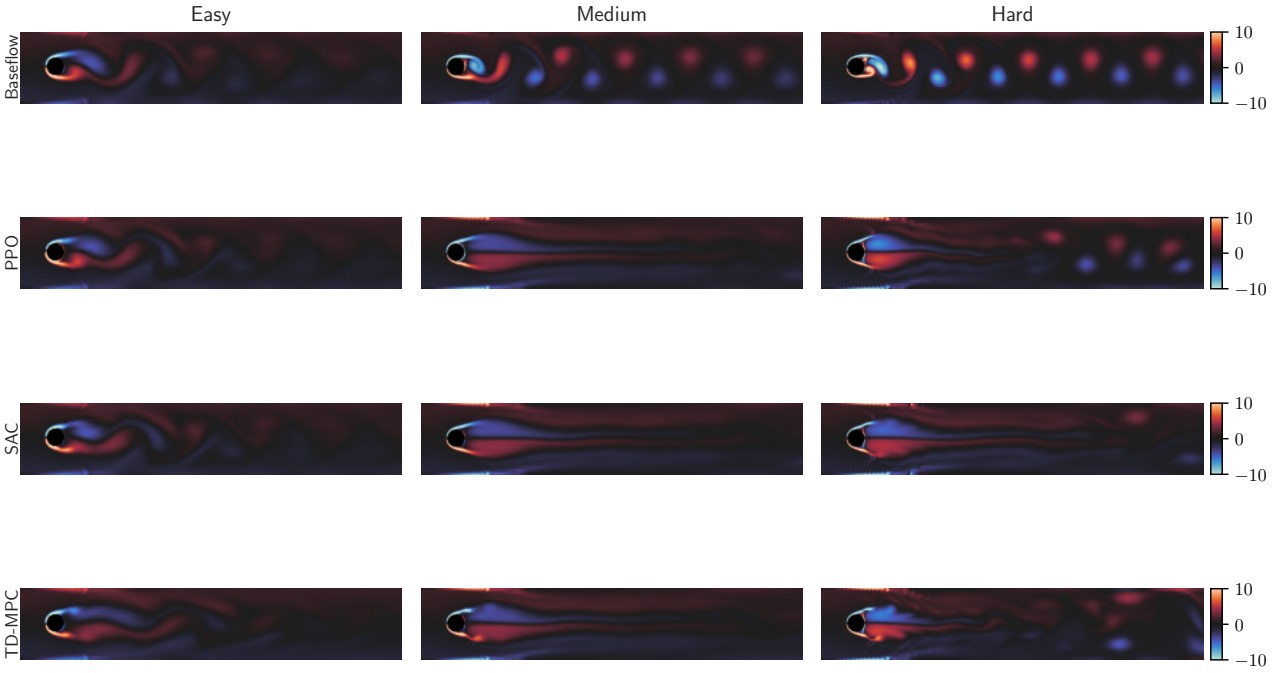

*Figure 37.* Qualitative test results for `CylinderRot2D`, showing the vorticity field.

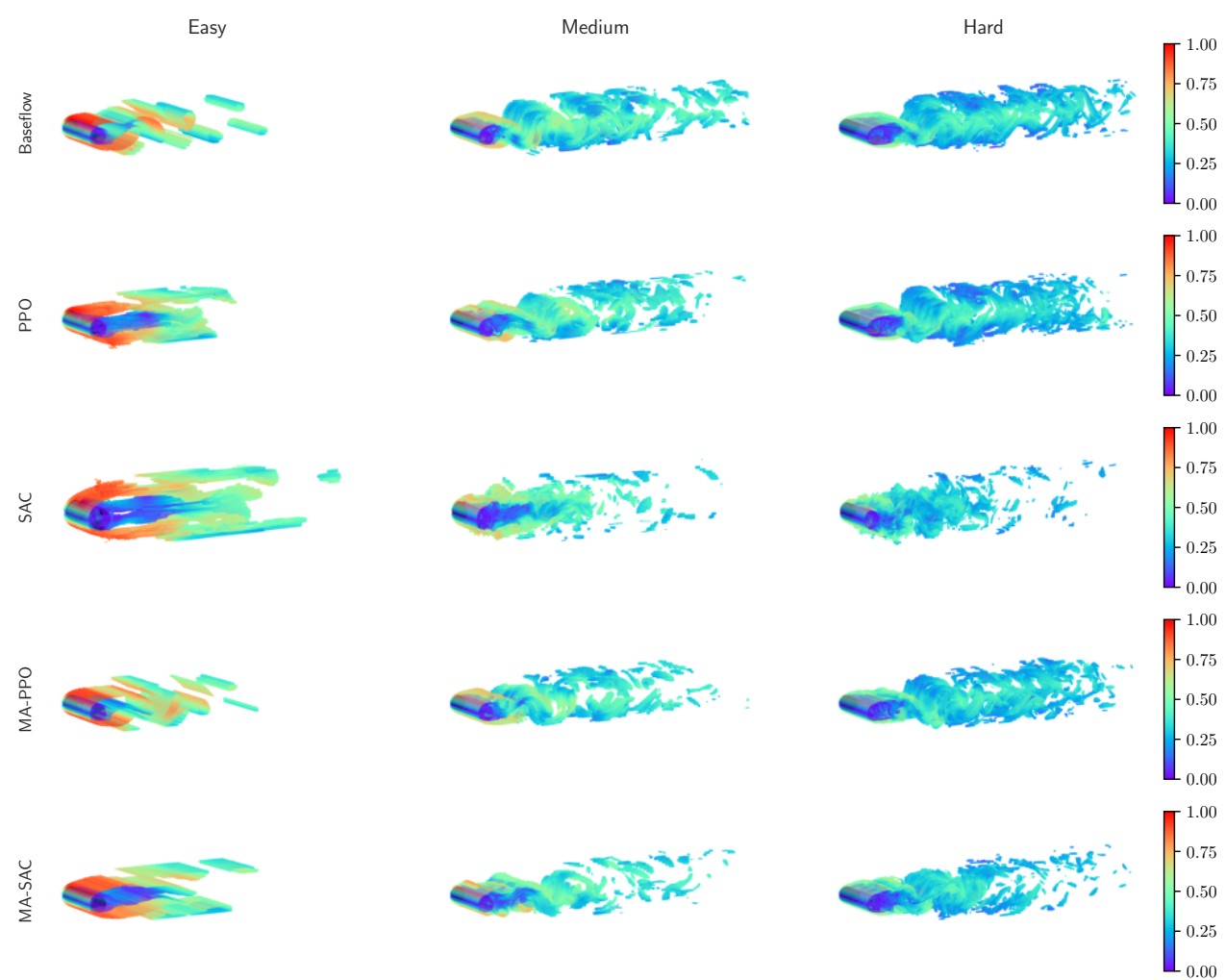

*Figure 38.* Qualitative test results for `CylinderJet3D`. Shown are isosurfaces of vorticity magnitude at levels 1.5, 2.5, and 3.5 for the easy, medium, and hard difficulty levels, respectively. Surface coloring indicates the velocity magnitude, normalized from $[0, 0.6u_{\text{mean}}]$ to $[0, 1]$ for each difficulty level individually. $u_{\text{mean}}$ refers to the mean temporal and spatial velocity of the uncontrolled flow.

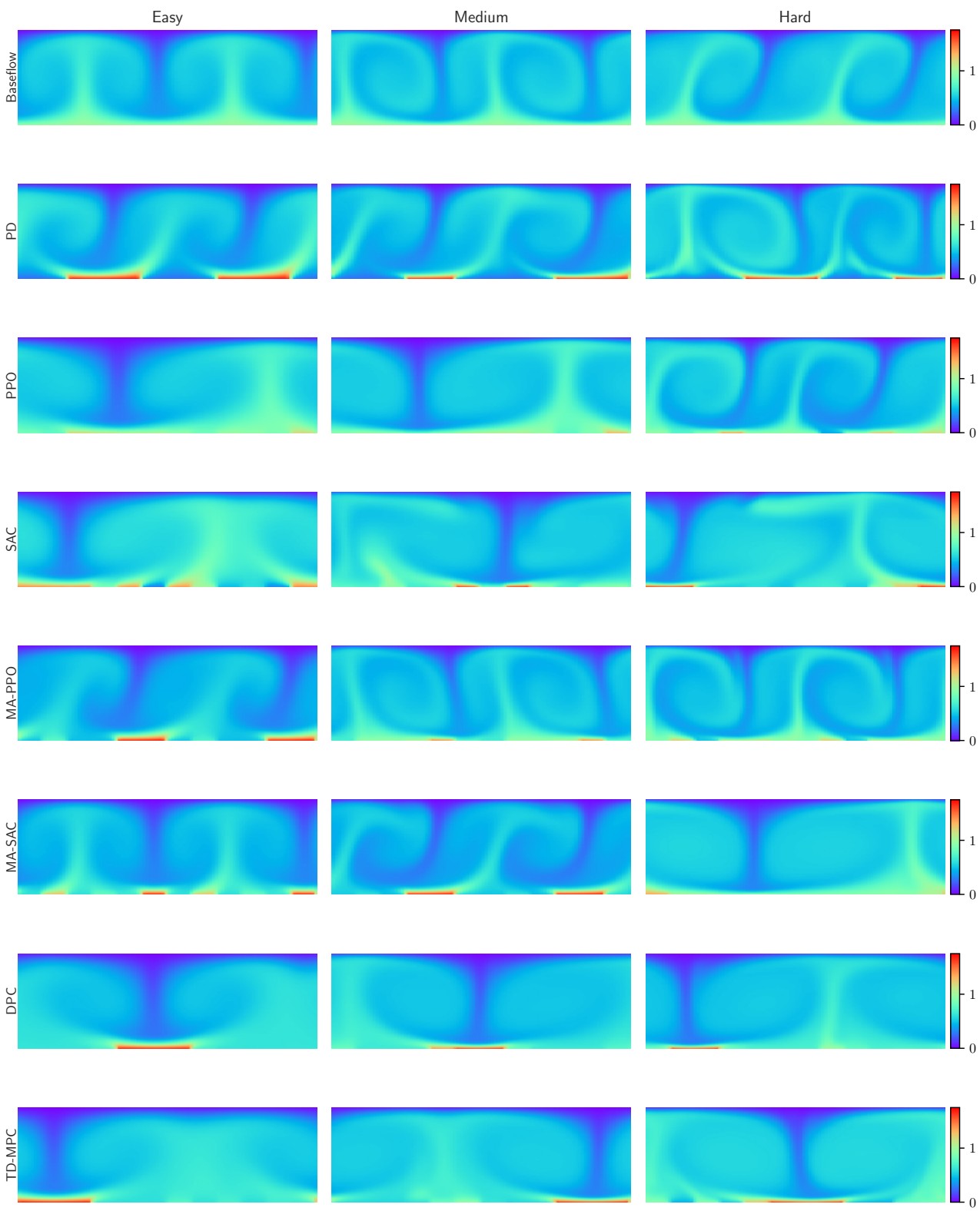

*Figure 39.* Qualitative test results for `RBC2D`. Coloring indicates temperature values ranging from 0 to 1.75.

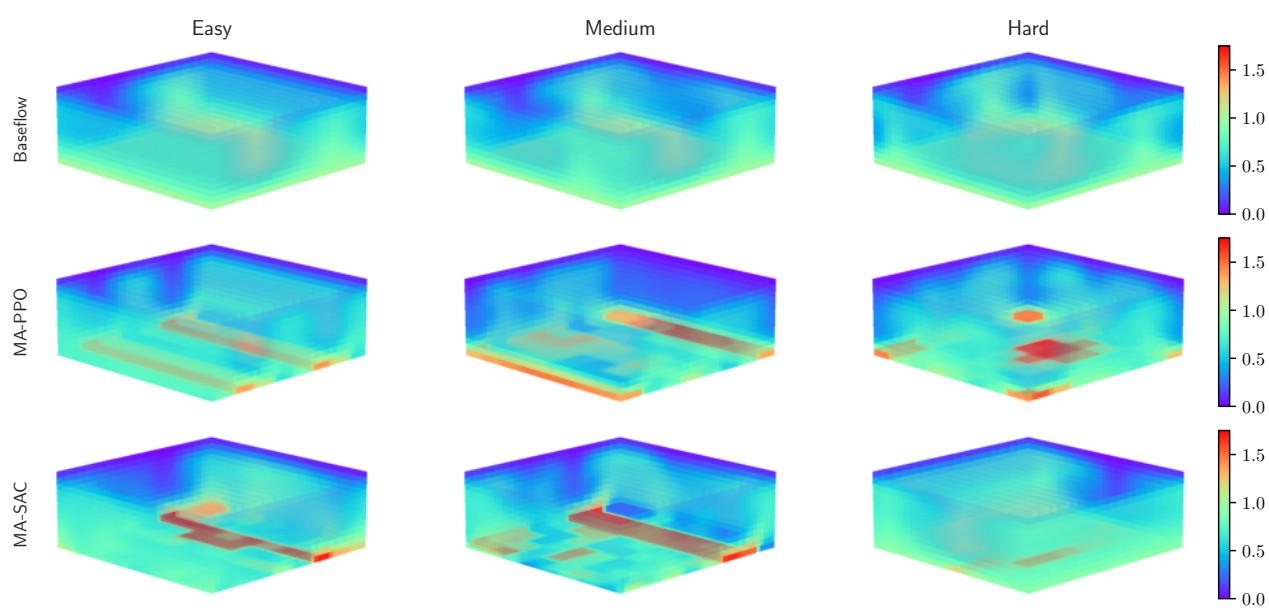

*Figure 40.* Qualitative test results for `RBC3D`. Coloring indicates temperature values ranging from 0 to 1.75.

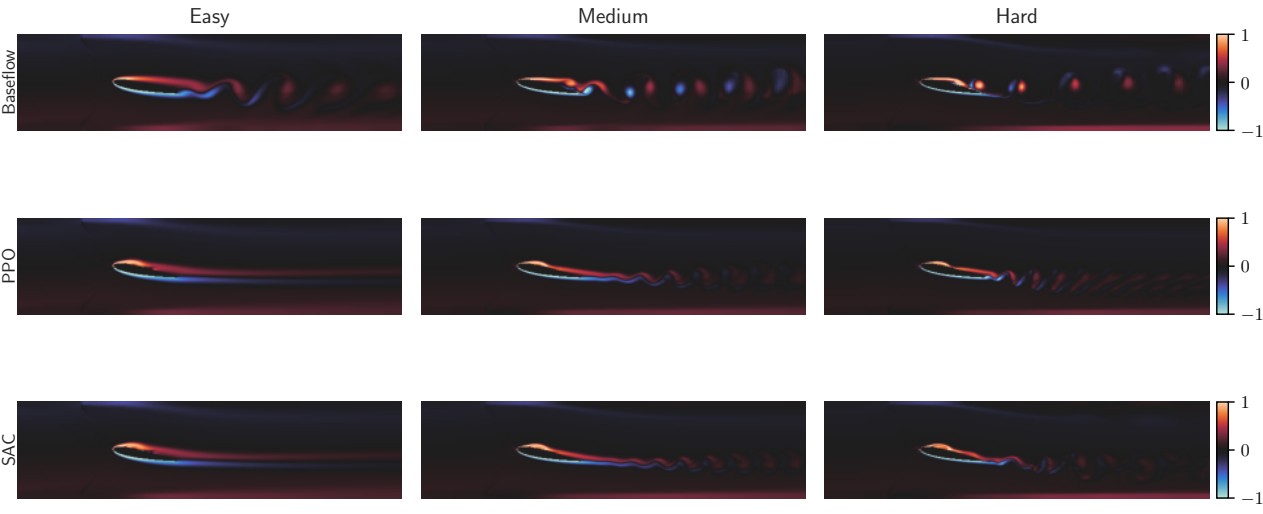

*Figure 41.* Qualitative test results for `Airfoil2D`. Visualizations show the vorticity normalized from $[-10, 10]$, $[-12.5, 12.5]$, and $[-15, 15]$ to $[-1, 1]$ for the easy, medium, and hard difficulty levels, respectively.

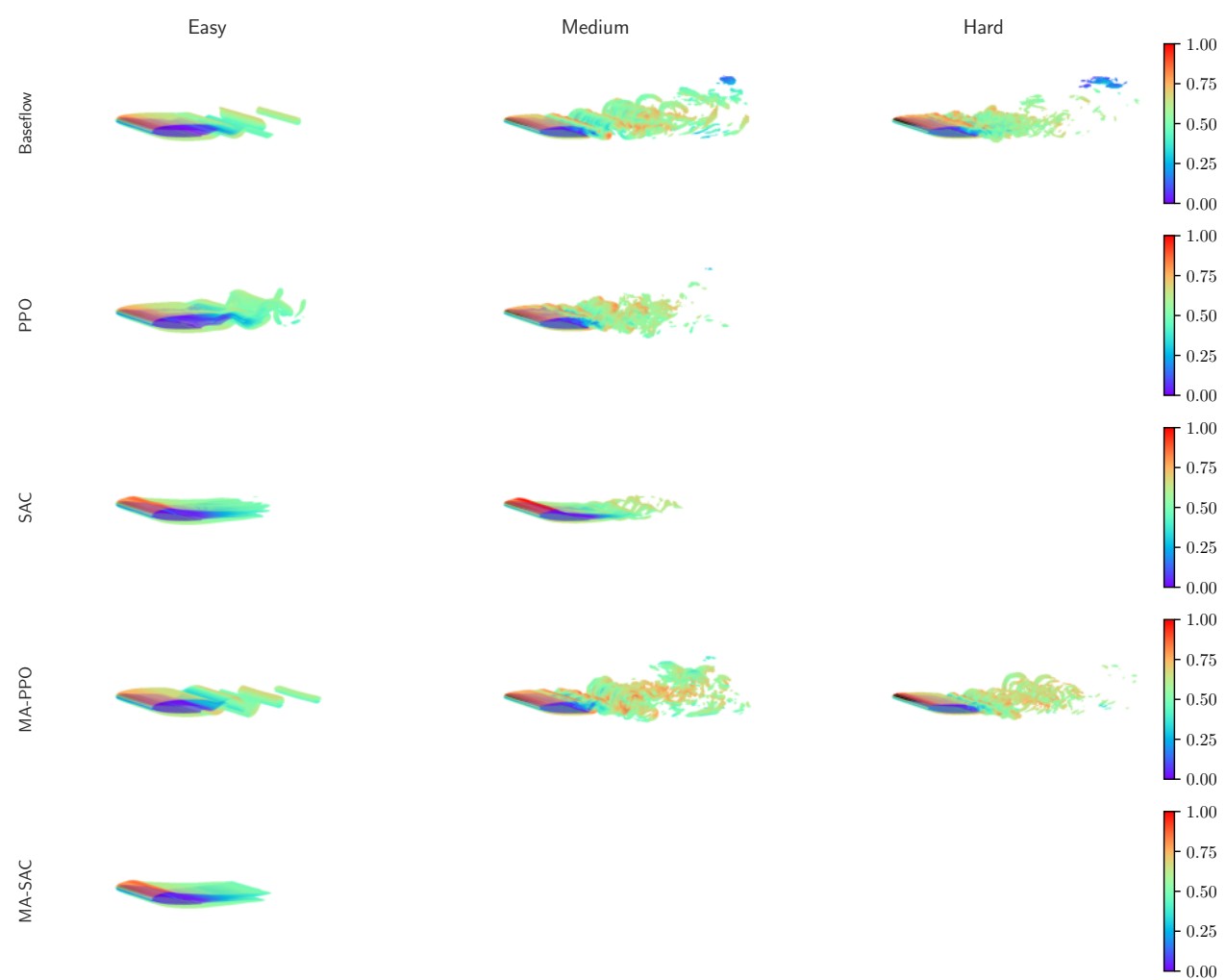

*Figure 42.* Qualitative test results for `Airfoil3D`. Shown are isosurfaces of vorticity magnitude at levels $2.0$, $3.5$, and $4.5$ for the easy, medium, and hard difficulty levels, respectively. Surface coloring indicates the velocity magnitude, normalized from $[0, 0.6u_{\mathrm{mean}}]$ to $[0, 1]$ for each difficulty level individually. $u_{\mathrm{mean}}$ refers to the mean temporal and spatial velocity of the uncontrolled flow.

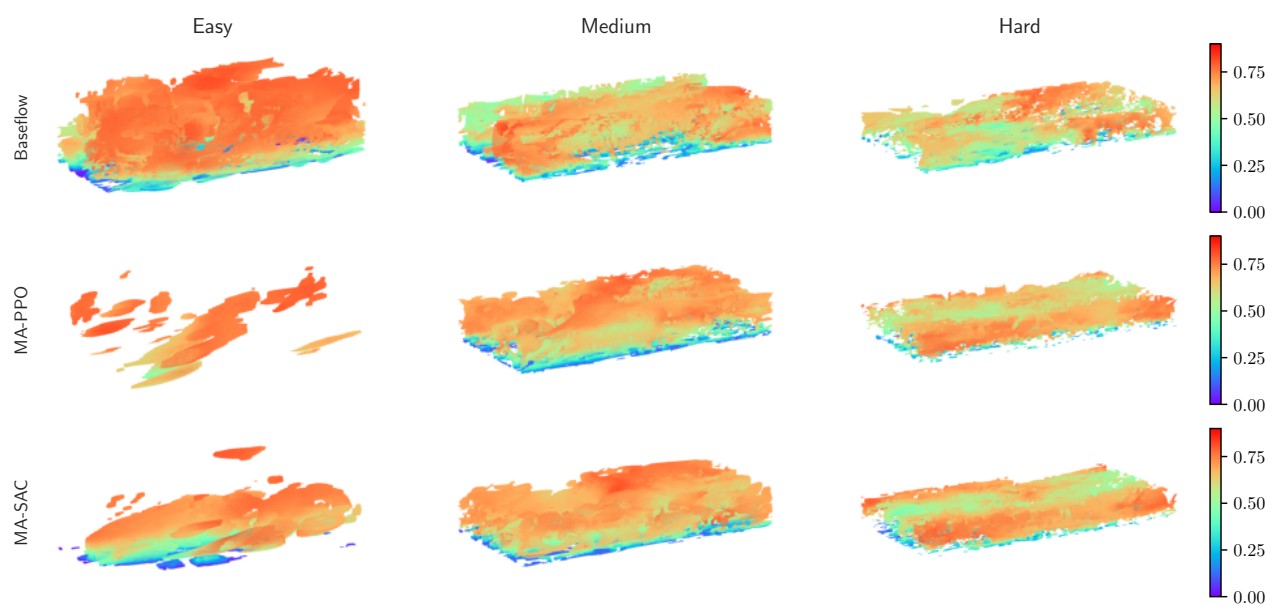

*Figure 43.* Qualitative test results for `TCFSmall3D-both`. Shown are isosurfaces of the Q-criterion (Jeong & Hussain, 1995) at level 0.05. Surface coloring indicates the velocity magnitude.

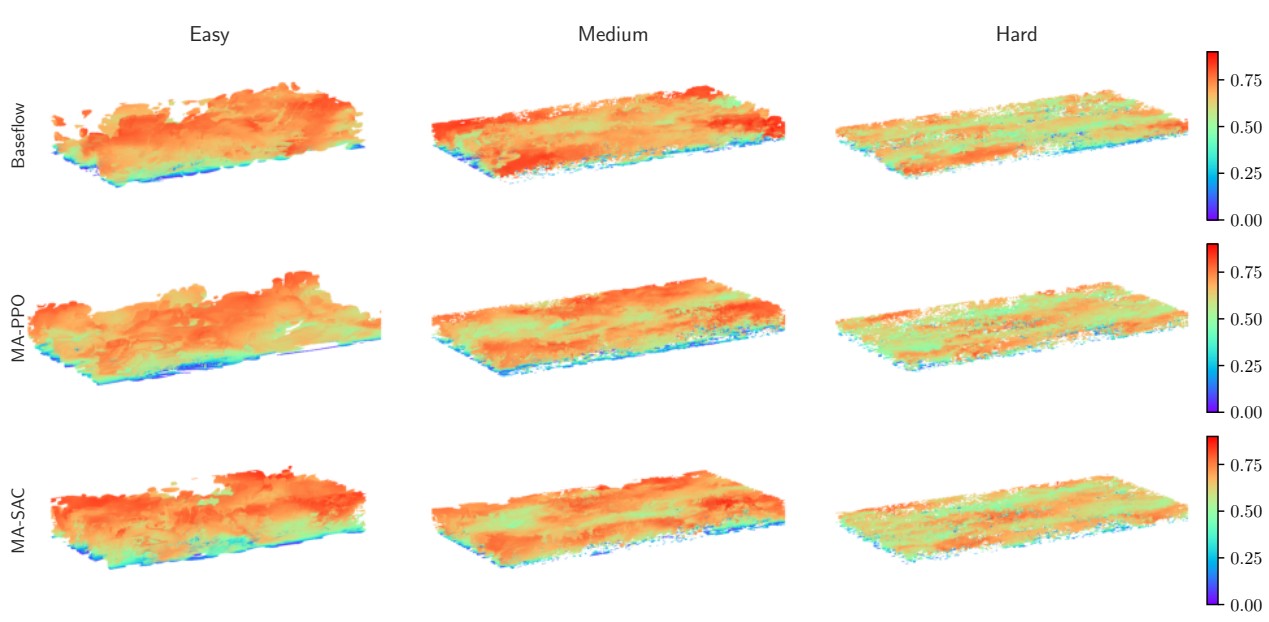

*Figure 44.* Qualitative test results for `TCFLarge3D-both`. Shown are isosurfaces of the Q-criterion (Jeong & Hussain, 1995) at level 0.05. Surface coloring indicates the velocity magnitude.

### E.5. PPO Network Size Ablation

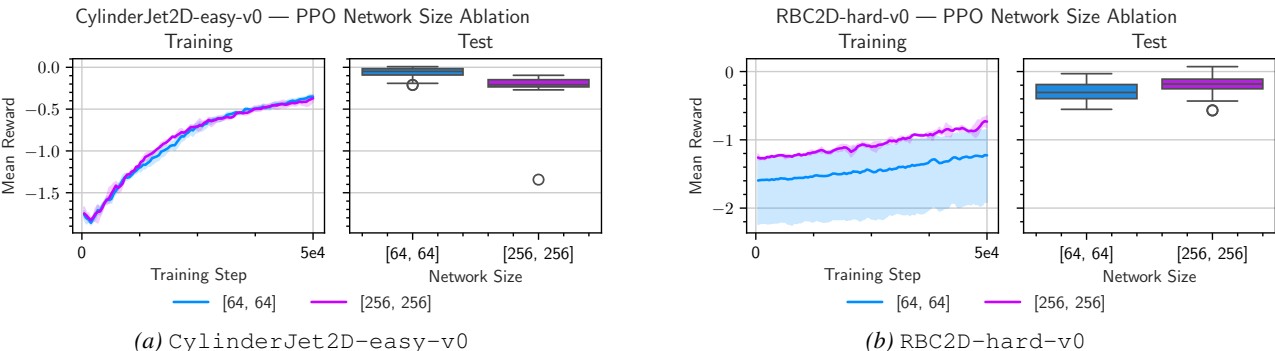

*(a)* `CylinderJet2D-easy-v0`   *(b)* `RBC2D-hard-v0`

*Figure 45.* Network size ablation for PPO across two environments. We evaluate hidden layer widths of 64 and 256 for the value and policy networks.

