# OpenReview forum: "Plug-and-Play Benchmarking of Reinforcement Learning Algorithms for Large-Scale Flow Control"
_ICML.cc/2026/Conference — ICML 2026 regular_

### Official Review · Reviewer_cgGu · 2026-03-09

**Soundness:** 2
**Presentation:** 4
**Significance:** 3
**Originality:** 3
**Overall Recommendation:** 5
**Confidence:** 4

**Summary:**

The paper makes a meaningful benchmark contribution for reinforcement learning in active flow control. FluidGym is presented as a standalone, fully differentiable, PyTorch-based benchmark that removes external CFD dependencies, standardizes training and evaluation, and releases environments, datasets, and an anonymized codebase.

**Compliance With Llm Reviewing Policy:**

Affirmed.

**Final Justification:**

The authors addressed my raised concerns and strengthened the paper. The paper makes a contribution towards efficient fluid dynamics learning control, analyzes numerical correctness, and provides evaluations across several algorithms.

**Key Questions For Authors:**

Please see suggestions for improvement above.

**Limitations:**

Yes

**Strengths And Weaknesses:**

**Strengths:**
- Larger-scale differentiable environments for (learning based) fluid control are a very relevant research direction.
- I appreciate the unusually thorough appendix, which includes solver validation, detailed environment definitions, training curves, hardware information, and hyperparameter settings.
- The paper is also refreshingly candid about current limitations, including limited statistical robustness due to the number of seeds, limited evaluation of differentiable control methods, and the use of default off-the-shelf hyperparameters.
- Overall, these make the work useful and likely valuable to the community.

**Weaknesses:**
- My main reservations are about empirical scope and presentation. The multi-agent setting is very relevant, and the paper does provide some motivation through spatially distributed control and transfer across domain sizes, but this case could be emphasized earlier and more clearly in the main text.
- I would also more explicitly highlight that these AFC problems are much more high-dimensional and nonlinear than conventional RL benchmarks, which has implications for architecture choice and algorithm design.
- As a benchmark paper, the work would be stronger with broader empirical coverage of RL/control methods and more direct numerical comparisons with prior AFC studies under matched settings.
- I would also be very interested in whether simpler discretized or parameter-shared multi-agent control baselines are competitive on some of these tasks (e.g., [1-2]).

- Finally, I think a few targeted ablations would strengthen the paper. The appendix indicates fixed PPO policy/critic widths of [64,64] across experiments; for some of the higher-dimensional domains, capacity and hyperparameter sensitivity studies would help clarify whether the reported results reflect algorithmic differences or mismatched default choices.
- The PICT solver validation is reassuring, but it would still be valuable to probe whether learned policies exploit any residual simulator inaccuracies, for example through additional stress tests or cross-validation under modified numerical settings.
- There is also an appendix inconsistency: Appendix C.4 lists the Airfoil hard setting at Re = 3×10^5, while Table 4 lists Re = 5×10^3; this should be corrected.

**References:**

[1] A. Tavakoli, et al. "Learning to Represent Action Values as a Hypergraph on the Action Vertices." ICLR, 2021.

[2] T. Seyde, et al. "Solving Continuous Control via Q-learning." ICLR, 2023.

---

> ### Author Rebuttal · Authors · 2026-03-30
>
> We thank the reviewer for the thorough feedback. Your suggestions helped us to significantly improve the paper by expanding the empirical scope. Key changes include:
> - **Empirical Breadth**: We have added Differentiable Predictive Control (DPC; Drgoňa et al., 2022) and TD-MPC (model-based RL) [1] as baselines, where DPC demonstrates strong sample-efficiency (factors of up to 10x for SAC and even up to 100x compared to PPO) enabled by end-to-end differentiability (see `dpc_and_td-mpc` in the repo).
> - **Evaluation of Simulator Inaccuracies**: To ensure policies do not exploit simulator inaccuracies, we conducted robustness tests under modified numerical settings (CFL condition), confirming the physical validity of learned policies.
> - **PPO Abalation**: We conducted a PPO network size ablation ([64,64] vs [256,256]) to ensure fair comparison with SAC.
> - **MARL & Challenges**: We moved the multi-agent motivation earlier in the text and expanded the discussion on the high-dimensional challenges of AFC.
> - **Corrections**: We fixed the Reynolds number inconsistency in Appendix C.4 regarding the Airfoil environment.
>
> Detailed responses to your points are stated below. Here, you find the additional plots and results: https://anonymous.4open.science/r/fluidgym-rebuttal.
>
> ## Weaknesses
> - **Emphasizing Multi-Agent Setting**: You are right that the multi-agent setting is a highly relevant feature of our benchmark. We agree that highlighting the motivation for MARL strengthens the paper. Therefore, we emphasize the motivation for MARL in Section 1, the MARL-native design desideratum of FluidGym in Section 3, and the relevance of our MARL results in Section 4, especially the transfer into larger, more complex domains.
> - **Highlighting Challenges of AFC**: Thank you for pointing this out. We have adapted the text in Sections 1 and 2 to emphasize the inherent challenges of AFC compared to conventional RL.
> - **Broader Empirical Coverage and Prior Comparisons**: To broaden our empirical coverage, we have expanded our baselines to include Differentiable Predictive Control (DPC), TD-MPC, and a classical PD controller (following previous work [2,3]). Regarding direct numerical comparisons with prior studies: We compared our results to previous work (cylinder drag reduction in agreement with Rabault, RBC with Vasanth, TCF with Guastoni). However, to emphasize this, we updated Section 4 of the manuscript to highlight the strong agreement with prior results previously stated in Appendix B.3.
> - **Discretized / Parameter-Shared MARL Baselines**: We would like to clarify that our current multi-agent baselines (MA-PPO and MA-SAC) already employ parameter sharing, as they train a single shared policy across spatially distributed pseudo-environments. However, we completely agree with the reviewer's underlying point regarding discretized action spaces and advanced Q-learning methods. The suggested works represent highly relevant methods for handling complex continuous or discretized control. While integrating these particular architectures is beyond the scope of this initial standard-baseline release, we have added a dedicated discussion section to the manuscript, which highlights them as promising candidates for future benchmarking within FluidGym.
> - **Ablations for Network Capacity**: We agree that the default [64, 64] network sizes of PPO might limit performance on higher-dimensional tasks and obscure algorithmic differences. To address this, we have added ablations of PPO using larger networks ([256, 256], same size as SAC) on the CylinderJet2D-easy-v0 and RBC2D-hard-v0 environments. The results suggest no significant impact on the algorithm performance in terms of training and test performance. We have included these results in the manuscript to clarify the sensitivity of PPO to network size (see `ppo_ablation` in the repo).
> - **Evaluation of Simulator Inaccuracies**: To ensure that learned policies are not simply exploiting numerical instabilities or residual inaccuracies, we conducted a sensitivity analysis. On the easy and hard RBC2D environments, we evaluated the SAC policies with a strictly modified numerical setting (CFL condition = 0.1). Both qualitative and quantitative results suggest that learned policies represent reasonable control behavior and do not exploit solver inaccuracies (see `stability_analysis` in the repo).
> - **Appendix Inconsistency**: We thank you for pointing this out. We have updated it with the correct value of 5x10^3.
>
> ## References
> [1] N. Hansen et al., “Temporal Difference Learning for Model Predictive Control,” ICML, 2022
>
> [2] G. Beintema et al., “Controlling Rayleigh–Bénard convection via reinforcement learning,” Journal of Turbulence, 2020
>
> [3] T. Markmann et al., “Control of Rayleigh-Bénard Convection: Effectiveness of Reinforcement Learning in the Turbulent Regime,” 2025, arXiv:2504.12000

---

> > ### Author Rebuttal · Reviewer_cgGu · 2026-04-03
> >
> > I thank the authors for their detailed responses. The additional analysis and discussion strengthen the paper and I'm happy to adjust the score.

---

> > > ### Author Response · Authors · 2026-04-07
> > >
> > > We would like to thank the reviewer for the positive feedback on our revisions, and for raising the score.

---

### Official Review · Reviewer_cvt6 · 2026-03-12

**Soundness:** 2
**Presentation:** 3
**Significance:** 2
**Originality:** 2
**Overall Recommendation:** 4
**Confidence:** 4

**Summary:**

The paper introduces a benchmark suite for active flow control that tries to clean up inconsistencies across prior numerical setups. The contribution includes end-to-end differentiable environments, multiple difficulty levels, and a 2D-to-3D transfer setting, with PPO and SAC used as baseline controllers. The paper also gestures toward coupling differentiable environments with standard RL methods, although the current experiments seem not to exploit that coupling as much as the framing suggests.

**Compliance With Llm Reviewing Policy:**

Affirmed.

**Key Questions For Authors:**

1. How exactly are the differentiable environments used during training, beyond standard simulator rollouts for PPO and SAC?
2. Were PPO and SAC tuned beyond default Stable-Baselines3 settings for these AFC tasks?
3. Can the authors report wall time and sample-efficiency comparisons between PPO and SAC, not just final scores? That would sharpen the practical value of the benchmark.

**Limitations:**

yes

**Strengths And Weaknesses:**

The motivation is solid, especially the attempt to standardize numerical setups and benchmarking practice. The main concern is the gap between framing and execution: the paper highlights differentiable environments and seems to suggest a tight coupling with PPO or SAC, but the experiments do not deliver that. I also worry that default Stable-Baselines3 settings may undersell the baselines, and the 2D-to-3D transfer setup seems constrained by the use of specific 2D sensor planes.

The paper would be stronger with explicit wall time comparisons between PPO and SAC. The related-work framing around prior AFC control cases could also be tightened with a few missing citations:
-  section 2 and table 1 : the beacon benchmark could be added to the table (Beacon, a Lightweight Deep Reinforcement Learning Benchmark Library for Flow Control)
- rayleigh-benard convection case : it would be fair to cite Beintema et al (Controlling Rayleigh–Bénard convection via reinforcement learning) as the first contribution using RL to control this case, even if the followed setup is that of Vignon et al.

---

> ### Author Rebuttal · Authors · 2026-03-30
>
> We thank the reviewer for the critical feedback. We have focused our rebuttal on closing the "gap between framing and execution" by adding new differentiable baselines and wall-clock time metrics.
>
> - **Tighter Coupling with Differentiability**: We have added Differentiable Predictive Control (DPC; Drgoňa et al., 2022) results. Unlike PPO/SAC, DPC explicitly exploits solver gradients, validating our differentiable framing with speedup factors up to 10x for SAC and even up to 100x compared to PPO.
> - **Baseline Coverage**: We added TD-MPC [1] as a model-based RL baseline as well as a PD-controller for RBC.
> - **Wall-Clock Time Reporting**: We now provide detailed wall-clock comparisons for all algorithms.
> - **Literature & Citations**: We updated the manuscript to include the Beacon benchmark and cite Beintema et al. as the pioneers of RL for RBC.
> - **Hyperparameter Ablations** for PPO show that performance differences are algorithmic rather than due to a lack of capacity.
>
> Detailed responses follow. New plots/results: https://anonymous.4open.science/r/fluidgym-rebuttal.
>
> ## Weaknesses
> - **Framing vs. Demonstration of Differentiability**: We agree that evaluating only PPO and SAC did not demonstrate the differentiable nature of our benchmark. To bridge this gap, we have implemented and evaluated DPC across 2D cylinder and RBC environments. By actively leveraging the gradients flowing through the solver, DPC shows strong sample efficiency, highlighting the practical value of the end-to-end differentiability (see `dpc_and_td-mpc` in the repo).
> - **2D-3D transfer**: We adopted specific sensor planes to maintain consistency with prior work and demonstrate that 2D-trained policies can generalize to 3D. This setup provides a computationally efficient proof-of-concept while leveraging FluidGym’s modular architecture for future implementation of more complex, sensor-invariant observation schemes.
> - **Wall-Clock Time Reporting**: We have added wall-clock evaluations for PPO, SAC, TD-MPC, and DPC. Notably, the high sample-efficiency of DPC (see above) comes at a wall-clock cost of 1.5–2x per step due to backward passes. While PPO and SAC show comparable runtimes, SAC is approximately 3x faster for massively parallel TCF environments (see `wallclock_times` in the repo).
> - **Missing Citations**: We appreciate the valuable literature suggestions. We added this sentence to the manuscript: "*Notably, Beintema et al. (2020) were the first to apply RL to control Rayleigh-Bénard convection and established the foundation for subsequent studies (Vignon et al., 2023)*".
> Also, we added this to Table 1 and updated the text accordingly:
>
> |Benchmark|No External Solver|Fully Differentiable|MARL|3D|
> |-|-|-|-|-|
> |Beacon (Viquerat et al., 2024)|Yes| No| No| No|
>
>
>
> ## Questions
> 1. **Usage of Differentiable Environments during Training**: You have raised a valid point. In the initial submission, the environment was largely treated as a black-box simulator for PPO/SAC. Therefore, in our newly added DPC baseline, the policies are explicitly - and exclusively - trained by exploiting the end-to-end differentiability of the benchmark: The simulator’s gradients are passed directly to the policy network via Backpropagation Through Time (BPTT). This tightly coupled optimization is what enables substantial improvements in sample efficiency compared to standard black-box rollouts in PPO, SAC, and TD-MPC.
> 2. **Hyperparameter Tuning of PPO/SAC**: Due to the inherent challenges of hyperparameter optimization, especially since even in similar domains, optimal hyperparameter configurations are not necessarily consistent (see [2] for instance), we use the default settings. However, as recently surveyed by Moslem et al. (2025), these are common settings for RL in AFC. We agree that hyperparameter tuning specifically for AFC is an important future direction. Our benchmark can serve as a first reference point by providing open-source, large-scale evaluations of default settings and enabling systematic comparisons between algorithms and hyperparameter configurations.
> To address concerns about the interplay of algorithmic performance and network sizes, we have added ablations of PPO using larger network architectures ([256, 256]) on the 2D Cylinder and RBC tasks (see `ppo_ablation` directory in the repo). The results so far suggest that the network size is not a key factor for performance differences between the benchmarked SAC and PPO configurations: a four times larger network exhibits very similar performance to our previous runs.
> 3. **Wall-Clock Time and Sample-Efficiency Comparisons**: We have added a figure and a table to the manuscript that highlight differences in algorithm runtimes across environment categories.
>
> ## References
> [1] N. Hansen et al., “Temporal Difference Learning for Model Predictive Control,” ICML, 2022
>
> [2] T. Eimer et al., “Hyperparameters in Reinforcement Learning and How To Tune Them,” in ICML 2023

---

> > ### Author Rebuttal · Reviewer_cvt6 · 2026-04-01
> >
> > I thank the authors for their rebuttal. All of my questions are now resolved, and as long as the two key points (differentiable algorithm and timings) are present in the final version, I will adjust my grade.
> >
> > Edit: I also want to point out that i was able to reproduce the results from the ablation using the provided codebase for the PPO_CylinderJet2D-easy. Timing were slightly higher but similar to what was announced in the main text, using one A100 GPU. Given the usual lack of reproducibility in DRL and in DRL+Physics, I thought this was an important thing to add.

---

> > > ### Author Response · Authors · 2026-04-07
> > >
> > > We would like to thank the reviewer for the positive feedback on our revisions and for the in-depth checking of our reported results! We are also grateful for receiving a higher score as a result.

---

### Official Review · Reviewer_HNH7 · 2026-03-13

**Soundness:** 3
**Presentation:** 4
**Significance:** 3
**Originality:** 2
**Overall Recommendation:** 5
**Confidence:** 4

**Summary:**

This paper introduces FluidGym, a benchmark suite for reinforcement learning in active flow control built around a differentiable PyTorch fluid solver. The benchmark covers cylinder, Rayleigh-Benard convection, airfoil, and turbulent channel-flow tasks, with 2D/3D settings, several difficulty levels, and both single-agent and multi-agent interfaces. The paper also reports baseline results for PPO, SAC, and their multi-agent variants, plus a small differentiable MPC demonstration.

**Compliance With Llm Reviewing Policy:**

Affirmed.

**Final Justification:**

This paper makes a useful benchmark contribution for reinforcement learning in active flow control. The main strengths are the standalone PyTorch based implementation, the breadth of the benchmark suite, and the clear presentation. My main concerns were about the novelty framing, the limited empirical support for the differentiability claims, and the strength of some comparative conclusions. The rebuttal addressed these concerns well through clearer positioning, additional differentiable and model based baselines, and more careful framing of the claims. Overall, I find the paper sound and valuable as a community resource, and the rebuttal positively changed my final recommendation.

**Key Questions For Authors:**

1. The paper's clearest novelty is full differentiability in a standalone single-stack benchmark, but the empirical study mostly evaluates model-free RL. Can you add a more systematic differentiable-control comparison, or otherwise weaken the framing around differentiability? A convincing answer here would materially improve my assessment of significance.

2. HydroGym is a close prior benchmark/resource paper in this area. Can you clarify more explicitly what is new here relative to HydroGym beyond implementation convenience and "all environments are differentiable"? A sharper positioning would directly affect my originality assessment.

3. The paper concludes that SAC reliably outperforms PPO, yet the study uses only default SB3 hyperparameters and only two baseline families. How robust is this conclusion to moderate hyperparameter tuning or to adding one or two stronger modern baselines? If the conclusion is not robust, I would prefer the claim to be weakened.

4. Several released variants are not benchmarked separately, the 3D airfoil experiments stop at the easy case, and some 3D settings use only three seeds. Which parts of the current experimental scope should readers treat as the core validated benchmark, and which parts should be viewed as preliminary? A clear answer would help calibrate the paper's claims.

**Limitations:**

Yes, these were discussed.

**Strengths And Weaknesses:**

### Strengths

- The resource itself looks useful. A single Python/PyTorch stack with no external CFD coupling is a real practical improvement over many existing AFC frameworks, and the paper is clear about why this matters for usability and maintenance.
- The benchmark design is broader than a minimal benchmark release. It spans four flow families, includes both 2D and 3D cases, supports SARL and MARL, and defines train/val/test splits with cached initial conditions rather than relying on ad hoc rollouts.
- The presentation is strong. I found the paper easy to follow, and the benchmark motivation, environment descriptions, and appendix organization are all above average for a systems/resource paper.
- The experimental reporting is more careful than in many RL-for-control papers. The paper uses 95% confidence intervals in the aggregate plots, IQM summaries in the appendix tables, reports seed counts, hardware/runtime details, and gives concrete PPO/SAC hyperparameters and policy architectures.
- I appreciate that the paper includes a direct limitations section and does not try to hide the practical constraints behind the current study.

### Weaknesses

- The main contribution is a benchmark/resource, not a new control method, and the paper should frame the novelty more narrowly. The closest prior work is HydroGym, which already offers a solver-independent flow-control RL platform with multiple validated environments and some differentiable support. The main novelty here is the cleaner fully differentiable, single-stack implementation across the released environments, not the broader idea of an RL benchmark for flow control.
- The differentiability angle is under-demonstrated relative to how central it is to the paper's framing. In practice, the paper mostly benchmarks standard model-free RL and only includes a limited D-MPC proof-of-concept on the cylinder setting. That is enough to show the feature exists, but not enough to establish that full differentiability materially changes what can be studied or achieved.
- The algorithmic conclusions are somewhat stronger than the evidence. The paper states that SAC reliably outperforms PPO across difficulty levels, but this conclusion is based on only two off-the-shelf baselines using default SB3 hyperparameters across highly heterogeneous tasks. For a benchmark paper, this is acceptable as an initial baseline release, but it is not strong enough to support a broad algorithmic takeaway without either tuning or a more diverse baseline set.
- Baseline coverage is still limited for a benchmark paper. Outside TCF opposition control and the small D-MPC example, I did not see stronger classical/model-based references that would anchor task difficulty, nor more recent RL baselines beyond PPO/SAC. This matters less for the benchmark release itself than for the comparative claims made around algorithm performance.
- The evaluation scope is somewhat narrower than the resource framing suggests. Some released environment variants are not benchmarked separately, the 3D airfoil study only includes the easy setting, and several expensive 3D cases are run with only three seeds. The statistical reporting is better than usual, but the low seed count still limits confidence in some of the finer comparisons.
- Related-work coverage is broadly reasonable, and I did not find a glaring missing benchmark citation. However, the contrast to HydroGym in particular should be sharper, because that prior work makes the originality here look more incremental than the current framing suggests.

---

> ### Author Rebuttal · Authors · 2026-03-30
>
> We thank the reviewer for the suggestions and positive feedback on our experimental reporting. Addressing your concerns regarding originality and demonstration of differentiability helped to improve the paper. Key changes:
> - **Differentiable Control Evaluation + Model-based RL**: We have added Differentiable Predictive Control (DPC; Drgoňa et al., 2022) and TD-MPC (model-based RL) [1]. DPC demonstrates strong sample efficiency (up to 10x vs. SAC and 100x vs. PPO) enabled by end-to-end differentiability.
> - **Framing of Novelty compared to HydroGym**: We clarified the novelty of FluidGym regarding the targeted audience and benchmarking standards.
> - **PPO Abalation**: We conducted a PPO network size ablation ([64,64] vs [256,256]) to ensure fair comparison with SAC.
>
> Additional plots can be found at https://anonymous.4open.science/r/fluidgym-rebuttal. Detailed responses to your points are stated below.
>
> ## Weaknesses
> - **Framing of Novelty (HydroGym)**: We agree and have narrowed the novelty framing. While flow-control RL benchmarks exist, our primary contribution is the fully differentiable, single-stack implementation across all environments.
> - **Demonstration of Differentiability**: To address this, we have expanded the evaluation to include DPC across CylinderJet2D and RBC2 environments. Results show strong sample efficiency, validating the potential of FluidGym’s end-to-end differentiability (see `dpc_and_td-mpc` in the repo).
> - **Robustness of Algorithmic Conclusions**: We acknowledge that default hyperparameters limit broad claims comparing algorithms and have weakend the conclusions in the manuscript. We have conducted ablations of PPO with larger networks ([256, 256]) suggesting that performance does not improve significantly with larger networks (see `ppo_ablation` in the repo).
> - **Baseline Coverage**: We have addressed the limited baseline set by adding DPC, TD-MPC (model-based RL), and a PD controller baseline for 2D RBC (see `dpc_and_td-mpc` in the repo). We have adapted comparative conclusions accordingly.
> - **Statistical Robustness**: We increased the number of random seeds from three to five for easy/medium 3D cylinder environments. Experiments for the remaining environments are in progress. Preliminary results show minimal variance (see `CylinderJet3D_increased_seeds` in the repo). The manuscript will be updated accordingly.
> - **Contrast to HydroGym**: Thank you for pointing this out. We clarify this in the response to the respective answer below and have updated the manuscript accordingly.
>
> ## Questions
> 1. **Systematic Differentiable Control Comparison**: The newly included DPC results provide the requested systematic comparison, demonstrating that gradient-based optimization significantly outperforms black-box RL in sample efficiency within our framework.
> 2. **Novelty compared to HydroGym**: Novelty in addition to the single-stack pipeline and native differentiability:
>     - **Target Audience**: HydroGym provides a very valuable platform coupling different CFD backends with RL interfaces. However, the setup and interaction with complex CFD codes make it hard for non-experts in fluid dynamics to enter the field. FluidGym, in contrast, is specifically tailored to the broader ML community due to its focus on ease-of-use. If researchers develop an RL method, a gradient-based control method, or a hybrid approach, they can benchmark it on all FluidGym test cases without thinking about the underlying CFD code.
>     - **Benchmarking Standards**: While HydroGym enables the evaluation of RL algorithms, the configuration of initial conditions, physical parameters, and evaluation protocols is left to the user. This inherently leads to incomparable results. FluidGym represents a fundamental shift from an environment wrapper to a comprehensive, standardized ML benchmark, following best practices for scientific benchmarks [2] such as independence of external data (here CFD) and distribution of baseline models alongside the benchmark
> 3. **Robustness of SAC vs. PPO**: Results from our network size ablations suggest the SAC performance advantage is relatively robust  (see `ppo_ablation` directory in the repo), but we will weaken the claim in the final text to specify that this ranking applies to the established baseline configurations.
> 4. **Core vs. Preliminary Benchmark**: We consider all presented environments as part of the core benchmark. Environments with the suffix `wide` are validated through their base environments and are included as additional domain-transfer challenges. We have added policy transfers to demonstrate the medium and hard 3D airfoil environments (see `airfoil_3d` in the repo). Baselines for all 3D environments with five seeds will be included in the final version.
>
> ## References
> [1] N. Hansen et al., “Temporal Difference Learning for Model Predictive Control,” ICML, 2022
>
> [2] J. Thiyagalingam, et al., “Scientific machine learning benchmarks,” Nat Rev Phys, 2022

---

> > ### Author Rebuttal · Reviewer_HNH7 · 2026-04-04
> >
> > Thank you to the authors for the detailed and constructive rebuttal. The additional experiments and clarifications have fully addressed my concerns and strengthened the paper. I have updated my recommendation accordingly.

---

> > > ### Author Response · Authors · 2026-04-07
> > >
> > > We would like to thank the reviewer for the positive feedback on our revisions, and for raising the score.

---

### Official Review · Reviewer_HLYB · 2026-03-16

**Soundness:** 3
**Presentation:** 3
**Significance:** 3
**Originality:** 3
**Overall Recommendation:** 4
**Confidence:** 3

**Summary:**

This paper is mainly a benchmark paper for reinforcement learning in active flow control. The authors argue that the field is hard to compare fairly because different papers use different CFD solvers, task setups, sensors, actuators, and evaluation protocols, so it is often unclear whether performance gains come from the algorithm or from the experimental setup. To address this, they introduce FluidGym, a unified benchmark suite built in PyTorch that does not rely on an external CFD solver, is fully differentiable, and supports not only standard 2D control problems but also larger 3D and multi-agent settings. They evaluate standard baselines such as PPO and SAC, and find that SAC is generally the stronger baseline overall. The main contribution is therefore not a new RL method, but a cleaner, more reproducible testbed that could make future research in flow-control RL easier to compare and build on.

**Compliance With Llm Reviewing Policy:**

Affirmed.

**Key Questions For Authors:**

1. How much of FluidGym’s claimed advantage comes from being standalone and PyTorch-native, versus from the benchmark design itself, and which of these do the authors expect will matter most for community adoption?
2. Since full differentiability is one of the paper’s main selling points, why does the evaluation focus mostly on PPO/SAC-style baselines rather than a broader set of differentiable control or model-based baselines?
3. Can the authors clarify the exact novelty claim behind “first standalone, fully differentiable benchmark suite,” especially in relation to recent benchmark platforms like HydroGym, which also position themselves as scalable RL platforms for fluid dynamics?
4. What is the authors’ intended scientific message: is the main takeaway benchmark standardization, evidence for differentiable control, or an empirical ranking of RL algorithms on AFC tasks?
5. Do the authors see FluidGym as primarily a benchmark for RL methods, or as a broader benchmark for AFC methods, and if it is the latter, why are classical and optimization-based controllers not compared more systematically across tasks?
6. see weaknesses

**Limitations:**

yes

**Strengths And Weaknesses:**

Strengths:
1. The paper tackles an important problem in active flow control research by providing a more standardized benchmark for fairer comparison across methods.
2. FluidGym is practically useful because it is standalone, PyTorch-based, and does not rely on an external CFD solver.
3. The benchmark is technically broad, with support for differentiability, single-agent and multi-agent RL, and both 2D and 3D environments.
4. The experiments are reasonably comprehensive for a benchmark paper, including multiple tasks, baseline comparisons, and transfer settings.

Weaknesses:
1. The paper does not compare against a sufficiently broad set of baselines, especially stronger differentiable control, model-based RL, or other methods that would better justify the value of a fully differentiable benchmark. Although differentiability is presented as a central advantage, the experiments mostly remain PPO/SAC benchmark comparisons, so the paper does not fully demonstrate the practical benefit of differentiable optimization in this framework.
2. The long-term impact of the work still depends heavily on whether the community actually adopts FluidGym as a standard benchmark.
3. The claim of novelty relative to prior benchmark platforms would benefit from tighter wording, since earlier systems such as HydroGym already addressed related benchmarking goals even if they did not offer the same standalone and fully differentiable setup.

---

> ### Author Rebuttal · Authors · 2026-03-30
>
> We thank the reviewer for the constructive feedback. Your comments helped us to significantly improve the paper’s empirical coverage and clarity on novelty. Key updates include:
> - **Expanded Baselines**: We have added Differentiable Predictive Control (DPC; Drgoňa et al., 2022) and TD-MPC (model-based RL) [1] as baselines. Notably, DPC demonstrates strong sample-efficiency (up to 10x vs. SAC and 100x vs. PPO), enabled by FluidGym’s end-to-end differentiability.
> - **Clarified Novelty**: We have sharpened the distinction from HydroGym regarding target audience, standardization, and accessibility.
> - **Scientific Message**: We have clarified that FluidGym is a standardized infrastructure for both RL and gradient-based AFC research.
> - **PPO Ablation**: PPO network size ablation to ensure fair comparison with SAC.
>
> Additional plots/results are at: https://anonymous.4open.science/r/fluidgym-rebuttal. Detailed responses to your points are stated below.
>
> ## Weaknesses
> 1. **Broadening Baselines & Differentiability**: We have expanded our baseline experiments to demonstrate the practical benefits of the end-to-end differentiability of our benchmark. We now include DPC and TD-MPC (model-based RL) as baselines. Results on CylinderJet2D and RBC2D show that leveraging FluidGym's gradients leads to 1–2 orders of magnitude faster convergence than derivative-free RL (see `dpc_and_td-mpc` directory in the repo). Furthermore, TD-MPC demonstrates the value of model-based RL for AFC.
> 2. **Community Adoption**: We believe the removal of external CFD dependencies can be a potential game changer for wide adoption, especially for non-experts in fluid dynamics. By abstracting the entire CFD stack, our benchmark can substantially lower the barrier to entry for RL researchers. Furthermore, the standardized benchmark design and publicly available baseline models allow researchers to compare methods without re-evaluating existing baselines.
> 3. **Novelty vs. HydroGym**: While HydroGym provides a unified RL interface coupled with multiple CFD backends, FluidGym is a single-stack benchmark specifically designed for the ML community. Key differences:
>     - **Target Audience**: HydroGym provides a very valuable platform for coupling different CFD backends with RL interfaces. However, the setup and interaction with complex CFD codes make it hard for non-experts in fluid dynamics to enter the field. FluidGym, in contrast, is specifically tailored to the broader ML community due to its focus on ease-of-use. If researchers develop an RL method, a gradient-based control method, or a hybrid approach, they can benchmark it on all FluidGym test cases without thinking about the underlying CFD code.
>     - **Standardization**: In previous frameworks, users must define the physical regimes, initial conditions, and evaluation protocols. To overcome this, FluidGym provides fixed train/test/val splits and evaluation protocols to ensure scientific rigor and reproducibility.
>     - **Accessibility**: By eliminating the need for CFD expertise, FluidGym allows researchers to benchmark RL, gradient-based, and hybrid methods via a simple pip installation.
> We will update the related work section to clarify this novelty.
>
> ## Questions
> 1. **Standalone vs. Benchmark Design**: These serve complementary goals. The standalone nature enables immediate accessibility and lowers the entry barrier. The benchmark design (standardized setups/splits/protocols) ensures scientific rigor and long-term comparability across the community. Both are important, but on different time scales.
> 2. **Focus on PPO/SAC**: As surveyed by Moslem et al. (2025), PPO and SAC remain the two most commonly used RL algorithms for AFC. To establish a robust baseline and compare our results to previous work, we initially focused on these two algorithms. We agree that model-based RL and differentiable control methods are important baselines and added DPC and TD-MPC results to bridge this gap. The DPC results highlight the benefits of a fully differentiable benchmark (see above). While differentiable RL for AFC is not yet widely explored, we believe our benchmark provides the necessary infrastructure to enable this research direction.
> 3. **Novelty vs. HydroGym**: See Weakness 3.
> 4. **Scientific Message**: The main takeaway is that our benchmark provides the standardized infrastructure that is required for the development and evaluation of AFC methods. It serves both as a benchmark for current RL methods and an enabler for research into differentiable control/RL.
> 5. **RL vs. General AFC Benchmark**: While primarily an RL benchmark, we view FluidGym as a broader testbed for AFC methods. To extend our existing study, we added and evaluated a PD controller baseline for the 2D RBC environments. While this is not the main focus of the benchmark, we agree that classical controllers can serve as baselines.
>
> ## References
> [1] N. Hansen et al., “Temporal Difference Learning for Model Predictive Control” ICML, 2022

---

> > ### Author Rebuttal · Reviewer_HLYB · 2026-04-06
> >
> > My concers have been fully resolved so I keep my postive score.

---

> > > ### Author Response · Authors · 2026-04-07
> > >
> > > We would like to thank the reviewer for the positive feedback on our revisions.

---

### Decision · Program_Chairs · 2026-04-30

**Decision:**

Accept (regular)

**Comment:**

This paper introduces a useful benchmark suite in PyTorch for active flow control. Reviewers initially scored between weak accept and accept showing strong consensus rather than disagreement, with the main takeaway being that a standalone single-stack pipeline is a signficant contribution to the community. The authors provided an effective rebuttal that strenghtened the paper considerably. They added Differentiable Predictive Control and TD-MPC baselines to properly demonstrate the benefits of the differentiability claim. They also included wall-clock time comparisons and PPO network ablations. Reviewers agreed these additions resolved their primary concerns. The manuscript is now well positioned to help standardize research in this space.